# Partial removal of visceral epididymal white adipose tissue in obese Ldlr-/-.Leiden mice impacts adipokine secretion, plasma free fatty acids, and improves cerebrovascular health

Florine Seidel[1,2], Martine C. Morrison[2], Ilse Arnoldussen[1], Vivienne Verweij[1], Simon Ebert[3], Joline Attema[2], Christa de Ruiter[2], Wim van Duyvenvoorde[2], Jessica Snabel[2], Bram Geenen[1], Ayla Franco[1], Eveline Gart[2], Jürgen Bernhagen[3,4,5], Maximilian Wiesmann[1], Robert Kleemann[2ↄ], Amanda J. Kiliaan[1ↄ*]

1 Department Medical Imaging, Anatomy, Preclinical Imaging Center, Radboud Alzheimer Center, Donders Institute for Brain, Cognition, and Behavior, Radboud university medical center, Nijmegen, The Netherlands, 2 Department of Metabolic Health Research, Netherlands Organisation for Applied Scientific Research (TNO), Leiden, The Netherlands, 3 Division of Vascular Biology, Institute for Stroke and Dementia Research (ISD), LMU Klinikum, Ludwig Maximilian University (LMU) Munich, Munich, Germany, 4 German Center for Cardiovascular Research (DZHK), partner site Munich Heart Alliance, Munich, Germany, 5 Munich Cluster for Systems Neurology (SyNergy), Munich, Germany

ↄ These authors contributed equally to this work.
* amanda.kiliaan@radboudumc.nl

## Abstract

Visceral white adipose tissue (WAT) dysfunction may contribute to obesity-related brain impairments but causal relationship has not been demonstrated. We herein investigated the impact of visceral epididymal WAT (eWAT) lipectomy on brain health and obesity-associated comorbidities (liver steatosis, atherosclerosis, WAT dysfunction) in obese Ldlr-/-.Leiden mice. High-fat diet (HFD)-fed obese mice underwent sham surgery or partial removal (~70%) of eWAT. A separate group of mice was kept on chow diet (control). Liver disease, atherosclerosis and three WAT depots were examined histologically, and WAT biopsies were also cultured *ex vivo*. Brain structure and function were monitored longitudinally using cognitive tests and neuroimaging, paralleled by histological analysis of brain pathology and hippocampal RNA-sequencing. In *ex vivo* WAT culture, the surgically removed eWAT portion secreted many adipokines and pro-inflammatory factors. Histological analyses at the end of the study showed that eWAT-lipectomy did not affect liver disease and atherosclerosis development, but reduced the number of severely hypertrophic adipocytes in the residual-eWAT. This was consistent with reduced secretion of adipokines (e.g., leptin, adiponectin) and pro-inflammatory mediators (e.g., PAI-1, MIP-1α/CCL3, IL-17) from the residual-eWAT in the *ex vivo* culturing experiments. Importantly, lipectomy alleviated HFD-induced adverse effects on hippocampal vasoreactivity, increased cortico-hippocampal (resting-state) functional connectivity and prevented the development of sedentary behavior. Lipectomy did not significantly affect histological

**Data availability statement:** The datasets are publicly available in figshare repository (DOI: https://doi.org/10.6084/m9.figshare.29520716. v4). The transcriptomics dataset generated for this study is publicly available in the Gene Expression Omnibus (GEO) repository (#GSE300930, https://www.ncbi.nlm.nih.gov/gds).

**Funding:** The work was supported by internal research programs of TNO, "ERP Body brain interactions" and "PMC Brain Health". In addition, J.B. acknowledges support from Deutsche Forschungsgemeinschaft (DFG) grant SFB1123-A2 and -A3 and from DFG under Germany's Excellence Strategy within the framework of the Munich Cluster for Systems Neurology (EXC 2145 SyNergy—ID 390857198).

**Competing interests:** The authors have declared that no competing interests exist.

**Abbreviations:** CBF: cerebral blood flow, CLS: crown-like structures, DCX: doublecortin, DHA: docosahexaenoic acid, DTI: diffusion tensor imaging, DVC: Digital Ventilated Cages, GLM: general linear model, GLUT-1: glucose transporter 1, FFA: free fatty acid, FGF21: fibroblast growth factor 2, HFD: high-fat diet, eWAT: epididymal white adipose tissue, HPS: hematoxylin-phloxine-saffron, IBA-1: ionized calcium-binding adapter molecule 1, IFN-γ: interferon γ, IFNA2: interferon α2, IFNAR: interferon α/β receptor, IP-10/CXCL10: interferon γ-inducible protein 10, IRF: interferon regulatory factor, MCP-1/CCL2: monocyte chemoattractant protein 1, MIF: macrophage migration inhibitory factor, MIP-1α/CCL3: macrophage inflammatory protein 1α, MRI: magnetic resonance imaging, mWAT: mesenteric white adipose tissue, MWM: Morris Water Maze, ORT: Object Recognition Test, PAI-1: plasminogen activator inhibitor 1, PBS: phosphate-buffered saline, rMWM: reverse Morris Water Maze, rs-FC: resting-state functional connectivity, SAA: serum amyloid A, SBP: systolic blood pressure, STAT1: signal transducer and activator of transcription 1, sWAT: subcutaneous white adipose tissue, TNF-α: tumor necrosis factor α, UPS22: ubiquitin-specific peptidase 22, WAT: white adipose tissue.

neuroinflammation or circulating cytokines/chemokines, but increased specific free fatty acids (e.g., eicosatrienoic acid and docosahexaenoic acid, known to have anti-inflammatory and vaso-protective properties). Hence, partial eWAT lipectomy in mice with manifest obesity partly prevents hippocampal cerebrovascular disturbances, demonstrating a causal involvement of visceral WAT in obesity-associated brain impairments. The beneficial effects of eWAT lipectomy may, at least partly, be mediated by anti-inflammatory free fatty acids, and possible changes in release of adipokines and inflammatory mediators.

## Introduction

Obesity is a disorder associated with numerous comorbidities including cardiovascular and metabolic diseases, such as type-2 diabetes, metabolic dysfunction-associated liver disease (MASLD) and atherosclerosis [1,2]. Obesity is primarily characterized by weight gain and excessive accumulation of white adipose tissue (WAT) mass [3], which expands through hyperplasia (increase in adipocyte number) and hypertrophy (increase in adipocyte size) [4]. Dysfunctional and inflamed WAT, especially visceral WAT [5], is associated with increased release of pro-inflammatory adipokines, systemic low-grade inflammation, metabolic dysfunctions and (cardio) vascular disturbances [6–9]. More recently, obesity has been further shown to impact brain health, increasing the risk of a decline in cognitive function related to memory and executive functions [10]. Obesity has in particular been associated with reduced grey matter volume, cortical thinning and impaired cerebral blood flow (CBF) in multiple regions of the brain [11–13], as well as decreased white matter integrity, as assessed by diffusion tensor imaging (DTI) [14]. Obesity has also been shown to dysregulate (resting-state) functional connectivity of brain networks involved in cognitive function [15]. Despite the plethora of associations observed between obesity and neurodegeneration, the knowledge regarding underlying biological processes is still limited. Several studies have suggested that visceral WAT may mediate the detrimental effects of obesity on brain health [16]. In line with this, visceral WAT has been specifically associated with metabolic abnormalities and cerebrovascular disturbances [5,17], although previous studies have demonstrated that adipocyte hypertrophy and WAT inflammation can be more prevalent in subcutaneous WAT [17,18]. Adipocyte hypertrophy is associated with the secretion of adipokines and pro-inflammatory mediators that may interact with the brain [19,20]. However, the implied role of visceral WAT in obesity-associated changes in the brain mainly originates from associative studies and a causal role of visceral WAT has not been demonstrated yet.

In the present study, we have therefore examined the role of visceral epididymal WAT (eWAT) in the development of obesity-associated metabolic, vascular, and cerebral complications using obese Ldlr-/-.Leiden mice. When fed a high-fat diet (HFD) with macronutrient composition and cholesterol content similar to human diets, Ldlr-/-.Leiden mice exhibit an obese phenotype with insulin resistance, dyslipidemia (elevated LDL-cholesterol and triglycerides), hypertension and atherosclerosis [21–23],

all of which are typical comorbidities of patients with obesity [1,2]. The eWAT of obese male Ldlr-/-.Leiden mice is the first WAT depot that develops adipocyte hypertrophy and inflammation upon HFD-feeding [22] and has been associated with metabolic dysfunctions akin to humans [24,25]. During HFD feeding, these mice reportedly also develop neuroinflammation, elevated blood pressure, hippocampal atrophy and long-term spatial memory impairment [21,26,27]. In the present study, we investigated the effects of partial visceral eWAT removal prior to eWAT inflammation onset: More specifically, the surgical eWAT lipectomy was performed after 8 weeks of HFD feeding, which corresponds to the time point when Ldlr-/-.Leiden mice are obese and eWAT mass is maximally expanded while not yet inflamed [22]. While this study emphasizes the effects of partial eWAT lipectomy on the brain, we also examined putative effects on other comorbidities (i.e., liver disease, atherosclerosis), as well as inflammation, adipocyte size and *ex vivo* adipokine/cytokine secretion from the eWAT (i.e., lipectomized eWAT collected at the time of surgery and residual eWAT collected at the end of the study), subcutaneous (inguinal, sWAT), and mesenteric WAT (mWAT). The effects of partial eWAT lipectomy on brain structure and function were determined using brain magnetic resonance imaging (MRI, including hippocampal volume, cortical thickness, CBF and white matter integrity), and cognitive tests. The study demonstrates that visceral WAT plays a causal role in the development of cerebrovascular dysfunction in obesity, which may be partly attributed to changes in the secretion of pro-inflammatory mediators and adipokines. Our findings may contribute to the development of therapeutic strategies targeting visceral WAT dysfunction, to alleviate obesity-associated brain impairments.

## Methods

### 1. Animals, diets, and study design

All animal experiments were conducted according to European Union regulations on animal research and approved by an independent Animal Welfare Body (number TNO-458) and the Veterinary Authority of Radboud university medical center (number 2017-0063-004).

Male Ldlr-/-.Leiden (n = 49) were obtained from a pathogen-free breeding stock at TNO Metabolic Health Research (Leiden, the Netherlands) and housed in the Pre-clinical Imaging Center (PRIME, Radboud university medical center, Nijmegen, the Netherlands). Mice were group-housed (2–4 animals) in digital ventilated cages (DVC, Techniplast SPA, Buguggiate, Italy) in which home-cage activity was measured 24 hours/day as described previously [28,29]. DVC data were analyzed on cage level and expressed per mouse. Mice were kept in a conventional animal room (relative humidity 50–60%, temperature 21°C) on a 12-hour light/dark cycle. To ensure equal housing conditions, cages from different groups were randomly distributed across shelves. The animals received food and autoclaved water *ad libitum*. Until the start of the study, all mice received a maintenance chow diet (Sniff R/M-H diet, Sniff Spezialdiäten GmbH, Soest, Germany).

At the start of the study (t = 0 weeks), 10–12 week-old mice were matched into two groups based on body weight, blood glucose, plasma cholesterol and triglycerides: one group (Chow group, n = 15) remained on chow diet, while the other group (HFD group, n = 34) was fed an obesogenic energy-dense HFD (45% kcal fat from lard, 20% kcal casein and 35% kcal carbohydrates, D12451, Research Diets, New Brunswick, NJ, USA, [23]). At t = 8 weeks, HFD-fed mice were matched again for the aforementioned parameters: half of the HFD-fed mice underwent a surgical procedure to partially remove eWAT (HFD + WATx group, n = 17) while the other half underwent a sham surgery (HFD+sham, n = 17). In the HFD + WATx group, about 70% of eWAT was removed (partial lipectomy) as detailed below. Only the researcher who allocated the mice to the groups was aware of the group allocation, while the other researchers conducting the experiments remained blinded regarding the group assignments. Group sizes were based on power calculations for detection of a difference in the end-points with lowest effect size (Morris Water Maze test and CBF measurements in MRI, f = 0.10, α = 0.05, power = 0.8), yielding a required sample size of n = 14 per group. One animal per group was added to compensate for potential drop-outs, and in both HFD-fed groups, two additional mice were added to compensate for potential loss due to the surgical procedures or ectopic dermatitis that may occur in HFD-fed mice upon frequent handling.

Food intake (at cage level and normalized to the number of animals per cage) and body weight (individual) were monitored weekly throughout the study. All animals underwent physiological assessments (blood pressure measurements, blood samplings), cognitive tests and brain MRI prior to surgery, and 1 month and 4 months after surgery. An overview of the study design is provided in Fig 1A. At t = 28 weeks, the mice were terminated by cervical dislocation after transcardial perfusion with phosphate-buffered saline (PBS) at room temperature (~0.8 ml PBS/g body weight, 5 ml/min). Brains, eWAT, inguinal sWAT, mWAT, livers, and hearts including aortic root were collected. Of note, in case of the HFD + WATx group, the eWAT obtained from lipectomy was used for histological analysis and immediate *ex vivo* culturing experiments while the remainder of the eWAT pad (referred to as 'residual-eWAT') was collected at the end of the study. Three mice from HFD + WATx group and one mouse from HFD+sham group were terminated earlier in the study due to health concerns resulting from either fighting or surgery complications and were therefore excluded from all analyses after surgery. In addition, one mouse from the Chow group died during the final MRI procedure and is therefore missing in the final MRI data and *post-mortem* analyses. An overview of the sample size for each analysis is available in S1 Table in S3 File.

## 2. Surgical procedures

After 8 weeks of HFD feeding, Ldlr-/-.Leiden mice are already obese and the eWAT is maximally expanded (i.e., maximal weight reached) but not yet inflamed based on histology [22]. Thus, the surgical removal of eWAT in HFD + WATx mice at 8 weeks could be regarded as a 'preventive treatment'. On average, 1.4 g of eWAT was removed corresponding to an estimated removal of 70% of the eWAT pad. For this estimation, the total eWAT weight for each animal at the time of surgery was calculated using the relationship between body weight and eWAT weight based on a previous time course experiment [22]. As control, HFD+sham underwent a sham surgery repeating the same procedure as for eWAT lipectomy but leaving eWAT pads intact. The detailed surgery protocols are available in the Supplementary material3. Systolic blood pressure (SBP)

At t = 5 weeks and t = 26 weeks SBP was measured as previously described [28,30]. Briefly, SBP was measured in a computerized and warmed tail-cuff plethysmography device (IITC Life Scientific Instruments, Woodland Hill, CA, USA) during 3 trials. In a first test trial (10 measurements), the mice were habituated to the restrainer and these data were not included. SBP was there measured in 2 trials (10 measurements each) over two consecutive days and expressed as mean SBP (mm Hg). Measurements presenting motion artefacts were excluded (S1 Table in S3 File).

## 4. Plasma analyses

### 4.1 Blood glucose, plasma triglycerides and cholesterol.
Blood (~200 µl) was collected via the tail artery at t = 5 weeks, t = 14 weeks, and t = 27 weeks, after a 5-hour fast. Blood glucose was measured with a glucometer at the time of blood sampling. EDTA plasma samples were prepared by centrifugation (10 min, 6000 rpm). In fresh plasma, total triglycerides and cholesterol concentrations were measured using enzymatic assays (GPO-PAP and CHOD-POP, respectively; Roche Diagnostics, Almere, The Netherlands).

### 4.2 Plasma biomarkers.
Enzyme-linked immunosorbent assays (ELISAs) were performed to measure plasma concentrations of insulin (#90080, CrystalChem, Zaandam, the Netherlands) at t = 5 weeks, t = 14 weeks and t = 27 weeks and leptin, adiponectin, resistin, macrophage migration inhibitory factor (MIF) and plasminogen activator inhibitor 1 (PAI-1) at t = 27 weeks (#DY498−05, #DY1119, #DY1069, #DY1978, #DY3828−05 kits respectively, all R&D Systems, Minneapolis, MN, USA). At t = 27 weeks, plasma S100-B (#CSB-EL020643MO, CUSABIO, Houston, TX, USA), and serum amyloid A (SAA, #KMA002, ThermoFisher Scientific, Waltham, MA, USA) were also quantified. In addition, cytokines and chemokines were measured in plasma by multiplex analysis using a V-PLEX Custom Mouse Biomarkers set (MesoScale Discoveries [MSD], Maryland, USA) including 'Proinflammatory panel 1 (interferon (IFN)-ɣ, IL-1β, IL-2, IL-4, IL-6, KC/GRO/CXCL1, IL-10, tumor necrosis factor (TNF)-α) and 'Cytokine Panel 1' (monocyte chemoattractant (MCP)-1/CCL2, IL-33, IL-27p28/IL-30, IL-17A/F, IFN ɣ-inducible protein (IP)-10/CXCL10). Plates were read on a MESO QuickPlex SQ 120

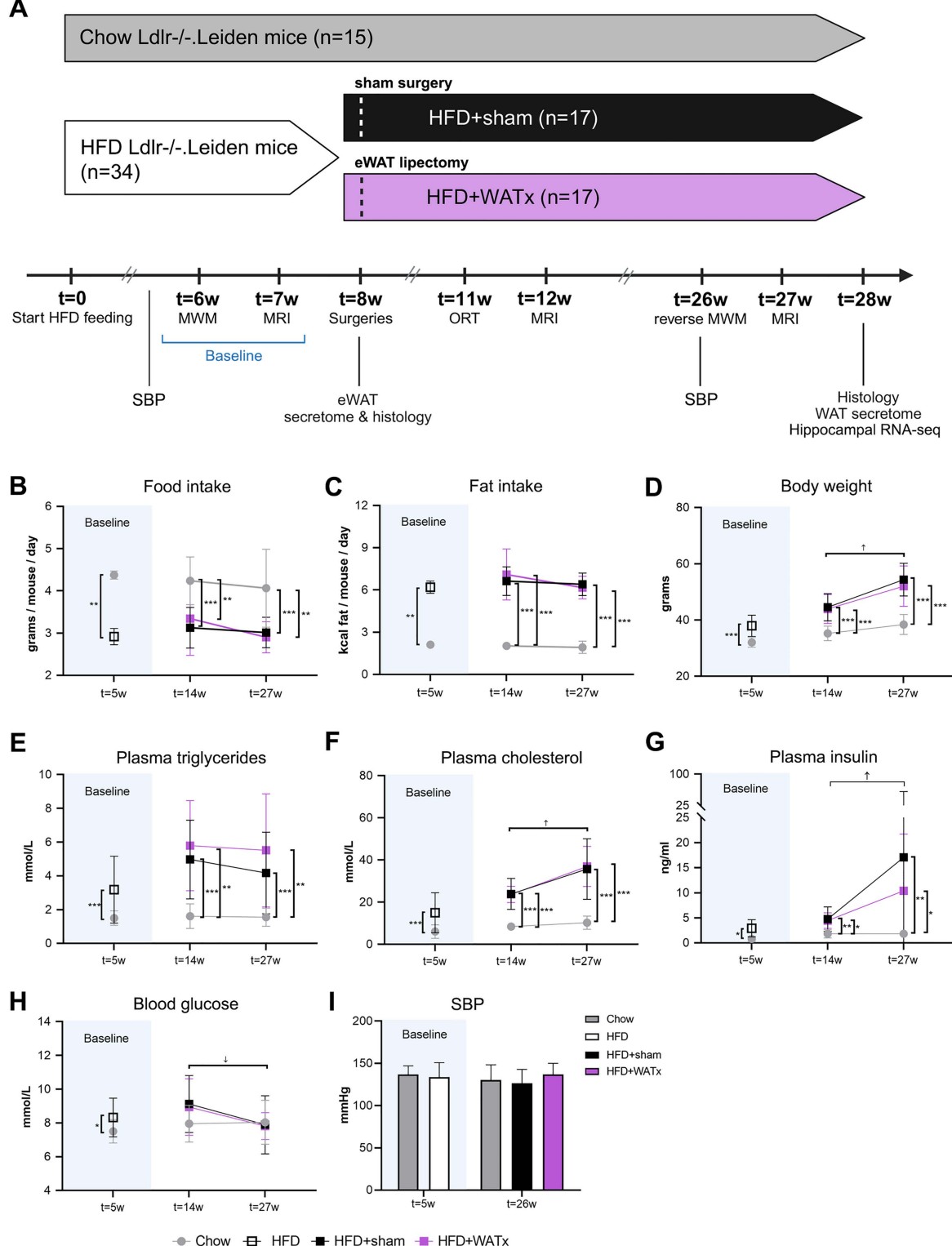

**Fig 1. Study design and general *in vivo* parameters.** (A) Study design: At t = 0, 10-12 week-old low density lipoprotein receptor-deficient substrain 'Leiden' (Ldlr-/-.Leiden) male were either fed a high-fat diet (HFD group, n = 34) or were kept on chow diet (Chow group, n = 15) as reference control. At t = 8 weeks, half of the HFD-treated mice underwent a sham surgery (HFD+sham group, n = 17) and the other half underwent partial (~70%) eWAT

lipectomy (HFD + WATx group, n = 17). Cognitive performance was assessed at t = 6 weeks (baseline; prior to surgery) with a Morris Water Maze (MWM) test, at t = 11 weeks (i.e., 1 month after surgery) with an Object Recognition Test (ORT) and at t = 26 weeks (i.e., 4 months after surgery) with a reverse MWM test. Systolic blood pressure (SBP) was measured at t = 5 weeks and t = 26 weeks. The mice also underwent brain magnetic resonance imaging (MRI) at t = 7 weeks (prior to surgery), t = 12 weeks (i.e., 1 month after surgery) and t = 27 weeks (i.e., 4 months after surgery) to study cortical thickness, hippocampal volume, grey and white matter integrity as well as brain perfusion and resting-state functional connectivity. Animals were housed in digital ventilated cages and home-cage activity was measured 24 hours/day. eWAT samples were collected during partial eWAT lipectomy for *ex-vivo* culture (secretome analysis) and histology. At the end of the study (i.e., t = 28 weeks), the (residual) eWAT, sWAT and mWAT depots of all mice were collected for *ex-vivo* culture and histology, and hippocampi were collected and used for RNAseq analyses to examine biological processes altered by HFD feeding and effects of partial eWAT lipectomy in the brain. (B) Food intake, (C) fat intake and (D) body weight were measured at t = 5 weeks (prior to surgery), t = 14 weeks (1 month after surgery) and t = 27 weeks (4 months after surgery). At the same time points, (E) triglyceride, (F) cholesterol and (G) insulin concentrations were measured in 5-hour fasted plasma, and (H) blood glucose levels were measured in whole blood. (I) Systolic blood pressure (SBP) was assessed at t = 5 weeks and t = 26 weeks. Data are shown as mean ± SD. ↑increase over time for all the groups (p ≤ 0.05), ↓decrease over time for all the groups (p ≤ 0.05). * p ≤ 0.05, ** p ≤ 0.01, *** p ≤ 0.001 between the experimental groups.

reader (MSD). In addition, free fatty acid (FFA) concentrations were determined in fasting plasma samples collected at t = 14 weeks using a targeted lipidomics approach on LC-MS/MS system as previously described in detail [31].

**4.3 Olink target 48 mouse panel analysis.** Protein expression was analyzed using the Olink Target 48 Mouse Panel (Olink Proteomics, Uppsala, Sweden), which enables multiplexed quantification of 45 mouse proteins. Mouse plasma samples were prepared according to the manufacturer's instructions. Briefly, 1 µL of each sample was incubated with paired oligonucleotide-labeled antibody probes. Upon binding to the target protein, the proximity of the probes enabled hybridization and extension by DNA polymerase, creating a unique PCR target sequence. Quantification was performed via high-throughput qPCR. Protein levels were expressed in Normalized protein expression (NPX) units on a log2 scale as well was the absolute concentrations. Data quality control and normalization were carried out using Olink's NPX Signature software (v2.0.2). Statistical analyses were performed using the NPX values. Proteins that were not detected are not shown.

## 5. WAT *ex vivo* culture

*Ex vivo* culture experiments were performed using WAT specimen collected at the end of the study (t = 28 weeks). In addition, in case of the HFD + WATx group, a portion of the surgically removed eWAT was cultured separately right after the lipectomy. ~100 mg of WAT were minced with sterile scissors and placed in 500 µl of medium (DMEM, high glucose, GlutaMAX™, pyruvate, #31966021 Gibco™, Thermofisher) containing 1% penicillin-streptomycin solution (#P0781, Sigma-Aldrich) and insulin (2 ng/ml, #I9278, Sigma-Aldrich). The tissues were incubated at 37°C (5% $CO_2$) and the culture media were collected after 4 hours. The media were centrifuged (13000 rpm, 2 minutes) and the concentrations of adiponectin, leptin, MIF, PAI-1 and resistin were measured in supernatants by ELISA as described for plasma experiments. KC/CXCL1, MCP-1/CCL2, IP-10/CXCL10, IL-10, macrophage inflammatory protein (MIP)-1α/CCL3, TNF-α, RANTES/CCL5 and IL-17 concentrations were analyzed using mouse cytokine and chemokine panels (Quanterix, Billerica, Massachusetts, USA) on an SP-X System as described [22]. The protein concentrations (per ml of medium) were normalized for the mass (gram) of eWAT being cultured during 4 hours. These concentrations were multiplied by the weight of the total eWAT depot to estimate the overall amount of adipokines/cytokines being released by the entire anatomical structure. The 4-hour secretion was expressed in ng or pg adipokines/cytokines being released. Of note, to calculate the secretome per total eWAT depot in the HFD + WATx mice at the time of lipectomy (t = 8 weeks), the total eWAT weight was estimated based on the earlier reported relationship between body weight and eWAT weight [22].

## 6. Brain MRI

At t = 7 weeks (i.e., prior to surgery), t = 12 weeks and t = 27 weeks (i.e., 1 month and 4 months after surgery), brain MRI was performed using a 11.7T BioSpec Avance III small animal MR system (Bruker Biospin, Ettlingen, Germany)

with Paravision 6.0.1 software (Bruker) as described previously [26,28,30]. Prior to MRI, mice were anesthetized with isoflurane (3.5% for induction, 1.8% for maintenance) in a 1:2 oxygen-medical air mixture. The MRI protocol and brain regions investigated are detailed in the Supplementary Material. Briefly, hippocampus volume and cortical thickness were measured in T2-weighted coronal images. White and grey matter integrity were assessed using DTI based on fractional anisotropy (FA) and mean diffusivity. CBF was assessed using an arterial spin labeling sequence under normal gas mix (1:2 oxygen – medical air), and again after switching to pure oxygen to induce vasoconstriction. Cerebral vasoreactivity, defined as the ability of the cerebral vasculature to adapt to vasoconstrictive conditions, was calculated by subtracting CBF measured under vasoconstrictive conditions from CBF measured under normal gas mix, divided by CBF measured under normal gas mix. Resting-state functional connectivity (rs-FC) between brain regions involved in diverse cognitive and motor processes was finally assessed using resting-state functional MRI acquisition.

## 7. Behavioral tests

To avoid habituation to a single behavioral test, different behavioral tests were performed at the different time points to assess spatial learning, memory, and explorative behavior. A t = 6 weeks, a Morris Water Maze (MWM) test was performed to evaluate spatial learning by measuring the time and distance to find a submerged platform (4-day learning phase, platform located in North-East quadrant of the pool), and short-term memory (probe test without the platform). At t = 11 weeks (i.e., 1 month after surgery), an Object Recognition Test (ORT) was used to assess explorative behavior and short-term memory by comparing the time spent exploring familiar and novel objects. A "discrimination index" (calculated as the difference between the time spent exploring the novel object and the time spent exploring the familiar object divided by the total exploration time) and a "recognition index" (calculated as the time spent exploring the novel object divided by the total exploration time) were determined. Lastly, at t = 26 weeks (i.e., 4 months after surgery), a reverse MWM (rMWM) test was performed to examine spatial learning and memory retention by relocating the former platform of the MWM, and conducting again learning and probe phases for a new platform location (platform located in South-West quadrant of the pool). All tests were analyzed using Ethovision XT software (v15, Noldus, Wageningen, the Netherlands). Details about the procedures are available in Supplementary Material.

## 8. Histological analyses

**8.1 Liver histology and atherosclerosis.** Liver steatosis, inflammation and fibrosis were determined by a board-certified pathologist on liver cross-sections (3 μm) stained with either hematoxylin-eosin or Sirius Red as described previously [32,33]. Atherosclerotic lesion size and severity were analyzed on cross-sections of the aortic roots (5 μm) stained with hematoxylin-phloxine-saffron (HPS) as reported previously [32].

**8.2 Brain (immuno)histopathology.** The staining procedures, antibodies used and quantification methods are detailed in Supplementary Material. Briefly, 4 free-floating (coronal) brain cross-section series were stained as follows: 1) one was stained for ionized calcium-binding adapter molecule 1 (IBA-1), a general marker for activated microglia, to assess neuroinflammation; 2) one for glial fibrillary acidic protein (GFAP), a marker to detect astrocytes, to assess astrogliosis; 3) one for glucose transporter 1 (GLUT-1), a protein expressed in the cerebral microvasculature, to assess vascular integrity; and 4) one for doublecortin (DCX), a marker for newly formed neurons, to assess neurogenesis. For IBA-1, GFAP and GLUT-1-stained slides (bregma ~ −1.94), the immuno-positive area and the number of positive particles (only for IBA-1) were determined in grey matter areas (cortex, hippocampus, thalamus) and white matter areas (corpus callosum, fimbria, optic tract, external capsule, internal capsule) using an automated quantification with ImageJ. For DCX-stained slides, DCX-positive cells were manually quantified in the dentate gyrus of the hippocampus.

**8.3 WAT inflammation and morphology.** At sacrifice, parts of residual-eWAT (left side), sWAT (left side) and mWAT were collected and fixed in paraformaldehyde and dehydrated overnight. At surgery, a portion of the removed eWAT

(left side) was processed the same way. Paraffin-embedded cross-sections (5 µm) were stained with HPS and scanned (Aperio Digital Pathology Slide Scanner AT2, Leica, Amsterdam, The Netherlands). WAT morphology was assessed quantifying adipocyte size in 3–5 non-overlapping fields using the Adiposoft plugin in ImageJ (v1.53, National Institutes of Health, United States) [34]. The proportions of very small (<2000 µm$^2$), small (2000–4000 µm$^2$), medium (4000–6000 µm$^2$), large (6000–8000 µm$^2$) and very large (>8000 µm$^2$) adipocytes were further quantified [30]. In the same fields, inflammation was determined as the number of crown-like structure (CLS).

## 9. Protein concentrations in brain cortex homogenates

Brain cortex homogenates were prepared in lysis buffer as previously described [27,35]. Concentrations of brain-derived neurotrophic factor (BDNF), IL-10, IL-15, IL-1β, IL-33, IL-6, MIP-1α/CCL3 and TNF-α were measured by multiplex analysis using a U-PLEX Custom Metabolic Group 1 set (Mesoscale discoveries [MSD], Maryland, USA). Protein levels were expressed per mg tissue.

## 10. Hippocampus gene expression by RNA sequencing

Hippocampi of the right hemispheres were isolated from fresh tissue, snap-frozen in liquid nitrogen and used for RNA isolation using the protocol previously described [36]. RNA concentrations were measured using a Nanodrop 1000 (Isogen Life Science, De Meern, the Netherlands) and RNA quality was assessed with a Bioanalyzer 2100 (Agilent Technologies, Amstelveen, the Netherlands). Next-generation RNA sequencing was performed at GenomeScan BV (Leiden, the Netherlands) and the obtained data were processed as reported previously [36]. Differentially expressed genes (DEGs) were determined with a statistical cut-off of p-value (p)<0.01 and used for gene enrichment analysis using Ingenuity Pathway Analysis suite (IPA; www.ingenuity.com, accessed on 2 October 2022). Canonical pathways and upstream regulators were considered significantly enriched when p < 0.01. When available, a z-score was calculated to indicate relevant inhibition or activation of the pathway or regulator. The transcriptomics dataset generated for this study is publicly available in the Gene Expression Omnibus (GEO) repository (#GSE300930, https://www.ncbi.nlm.nih.gov/gds).

## 11. Statistical analyses

All data are shown as mean ± standard deviation (SD). Statistical analyses were performed with SPSS software (v28, IBM, Armonk, NY, USA) and a cut-off of p ≤ 0.05 was used. The sample size for each analysis is reported in S1 Table in S3 File. Variables that were not normally distributed were transformed according to the Tukey ladder of powers. When transformation of the data was not possible, non-parametric tests (Kruskal-Wallis) were used. For parameters assessed at only one time point (e.g., measurements prior to surgery, *post-mortem* analyses), univariate/multivariate general linear models (GLM) with Bonferroni correction for multiple testing were performed with either Tukey (homoscedasticity) or Dunnett's T3 (heteroscedasticity) *post-hoc* tests. For parameters assessed at multiple time points (e.g., MRI measurements at t = 12 weeks and t = 27 weeks, (reverse) MWM acquisition phase), intra-group effects over time and overall inter-group effects were determined using repeated-measures GLM with Bonferroni correction followed by Tukey or Dunnett's T3 *post-hoc* tests. When significant time-by-group interaction was found, the dataset was split and the repeated-measures GLM was performed again to assess intra-group variations over time and multivariate GLM was used to assess inter-group differences at each time point separately. For CBF outcomes measured under normal gas mix, an additional repeated-measured GLM with Bonferroni correction was performed to assess CBF changes over time within the three experimental groups Chow, HFD+sham and HFD + WATx including t = 7 weeks, t = 12 weeks and t = 26 weeks timepoints. For this specific analysis at t = 7 weeks (prior to surgery), the combined HFD group was retrospectively divided into HFD+sham and HFD + WATx subgroups, mirroring the group separation used at the post-surgery time points.

## Results

### 1. Partial eWAT lipectomy does not alter HFD-induced obesity phenotype

During the whole study, food intake and dietary fat intake were similar in the HFD-fed groups (Fig 1B-C). Already after 5 weeks of HFD feeding (prior to surgery), and at all time points after surgery, HFD-fed mice exhibited an obesity pheno-type. Compared with the Chow group, the HFD+sham and HFD+WATx groups had a comparably higher body weight, and elevated plasma concentrations of cholesterol, triglycerides and insulin (Fig 1D-G). Blood glucose was higher in the HFD group compared with the Chow group at t = 5 weeks only (Fig 1H). No differences in SBP were observed between the groups (Fig 1I). HFD+sham and HFD+WATx comparably developed pronounced forms of liver steatosis, lobular inflam-mation and liver fibrosis as well as atherosclerosis in contrast to the Chow group which hardly developed pathology (S1 Fig in S2 File).

### 2. Partial eWAT lipectomy is associated with a lower number of very large adipocytes in the residual-eWAT

The average weight of the eWAT portion which was removed during lipectomy was 1.4 ± 0.4 g. Representative images of eWAT histology are provided in Fig 2A. At the end of the study, the remainder of the eWAT was significantly smaller in the HFD+WATx group (0.8 ± 0.2 g) compared with HFD+sham controls (2.2 ± 0.5 g, p < 0.001), and also than in the Chow reference (1.4 ± 0.7 g, p = 0.021, Fig 2B).

At the time of surgery, there was no inflammation in the removed (lipectomized) portion of eWAT (Fig 2C) and the average adipocyte size was already comparably high as in the eWAT of the HFD+sham mice at the end of the study (Fig 2D). At the end of the study, the number of crown-like structures (CLS) indicating inflammation was significantly higher in HFD+sham and HFD+WATx groups compared with Chow (p < 0.001 and p = 0.011 respectively), and the number of CLS was similar in HFD+sham and HFD+WATx groups (p = 0.720). The average adipocyte size was similar in the residual-eWAT of HFD+WATx mice and in the eWAT of HFD+sham mice (p = 0.400), but the percentage of very large adipocytes, considered to be important for the release of inflammatory molecules [19], was significantly lower in the residual-eWAT of the HFD+WATx group compared to the HFD+sham group (HFD+WATx 12.9 ± 5.0% vs HFD+sham 18.9 ± 5.0%, p = 0.042, Fig 2E).

Additionally, mWAT and sWAT were collected at the end of the study and analyzed histologically: the average adipocyte size in mWat and sWAT, and CLS counts in mWAT were increased by HFD feeding, and eWAT lipectomy did not affect HFD-induced changes in mWAT and sWAT with regards to inflammation and morphology (S2 Fig in S2 File).

### 3. Partial eWAT lipectomy partly modulates the secretome of *ex vivo* culture eWAT

**3.1 eWAT secretome at the time of surgery.** Tissue samples of the surgically removed eWAT were cultured *ex vivo* to gain insight in the type of inflammatory factors being released by the removed portion of adipose tissue. The amount of inflammatory mediators that were secreted during a 4-hour period of culturing were determined in the supernatant of cell culture medium. Culture medium from wells treated the same way but not containing eWAT served as 'blank' control samples. Obtained data were normalized to the total eWAT weight to gain insight into the 4-hour secretion capacity of the entire eWAT depot (Table 1). This analysis shows that, although the eWAT depot hardly contained CLS at the time of surgery, it already had the capacity to secrete multiple (pro-inflammatory) factors, and it suggests that, upon surgery, mice of the HFD+WATx group were less exposed to these factors derived from eWAT.

**3.2 eWAT secretome at the end of the study.** At sacrifice, the different WAT depots of all groups were collected and also cultured *ex vivo*. In case of the HFD+WATx group, the residual-eWAT was cultured. The results of a 4-hour secretome analysis of the eWATs are shown in Table 2. Compared with Chow, the eWAT of HFD+sham and HFD+WATx groups had a lower secretion capacity of resistin (HFD+sham vs Chow p = 0.032, HFD+WATx vs Chow p = 0.005), MCP-1/CCL2 (HFD+sham vs Chow p = 0.011, HFD+WATx vs Chow p = 0.003) and IP-10/CXCL10 (HFD+sham vs Chow p = 0.005,

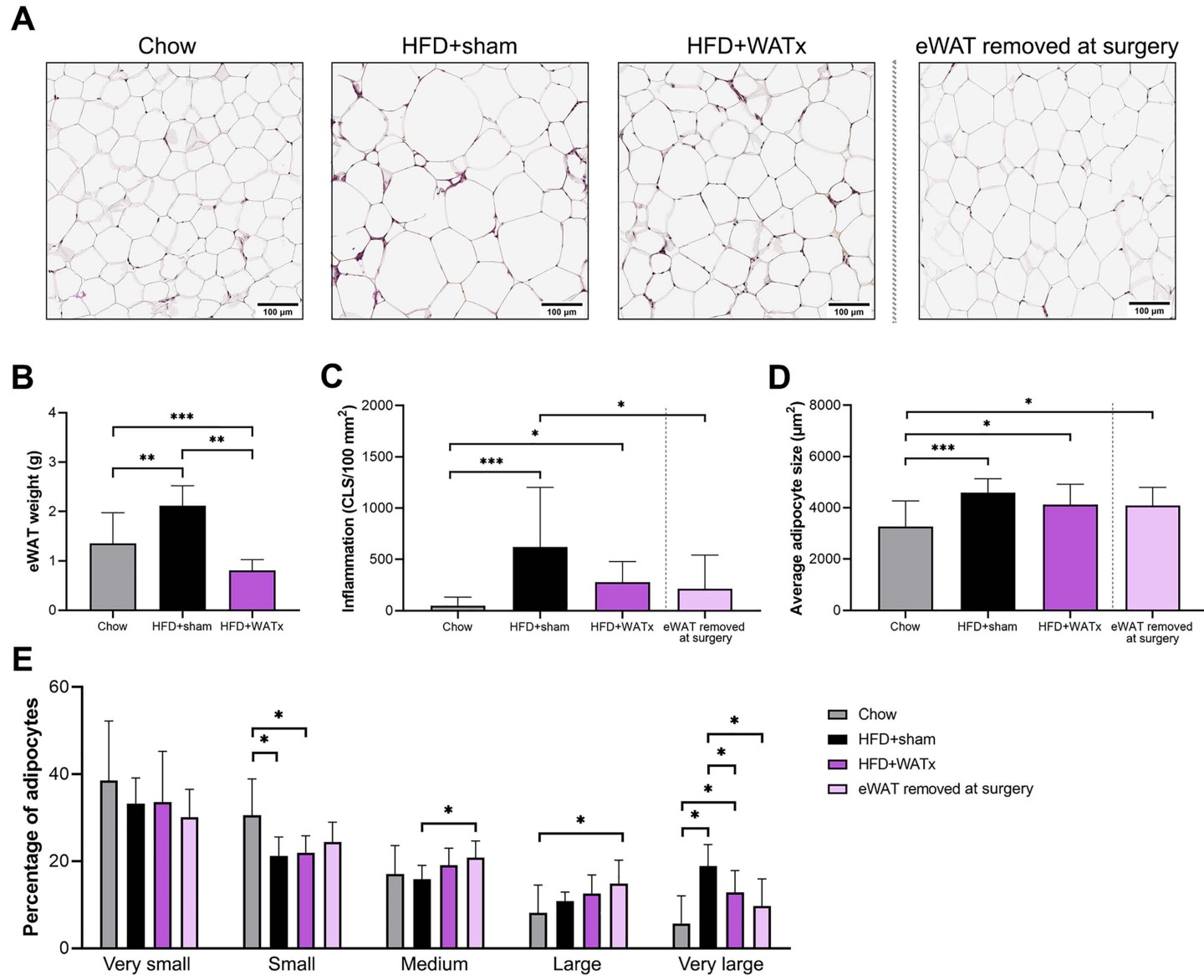

**Fig 2. Inflammation and morphology of the eWAT at the end of the study.** (A) Representative images of eWAT histology of Chow, HFD+sham and HFD + WATx groups at the end of the study, and of the surgically removed eWAT at the time of lipectomy (t = 8 weeks). (B) eWAT weight, (C) inflammation (crown-like structures, CLS) and (D) average adipocyte size. (E) Percentage of adipocytes per size category, very small: < 2000 μm², small: 2000-4000 μm², medium: 4000-6000 μm², large: 6000-8000 μm², very large: > 8000 μm². Data are shown as mean ± SD. * p ≤ 0.05, ** p ≤ 0.01, *** p ≤ 0.001 between the experimental groups.

HFD + WATx vs Chow p = 0.002). Notably, the secretion capacities of adiponectin, leptin and PAI-1 were significantly lower in HFD-WATx mice compared with HFD+sham mice (adiponectin p = 0.020, leptin p = 0.006 and PAI-1 p = 0.019), indicating that partial eWAT lipectomy affected these factors. The secretion capacities of MIP-1α/CCL3 and IL-17 were also numerically lower in HFD + WATx mice compared with HFD+sham mice, but the effects did not reach statistical significance (for MIP-1α/CCL3 HFD + WATx vs HFD+sham −86%, p = 0.105; for IL-17 HFD + WATx vs HFD+sham −60%, p = 0.055). No significant differences between HFD+sham and HFD + WATx were observed for mWAT (S2 Table in S3 File)

**Table 1. Inflammatory proteins secreted within 4 hours by the surgically removed eWAT.**

| Proteins | Secretion capacity of the removed eWAT depot |
|---|---|
| Adiponectin (ng) | 2948.0±782.4 |
| Resistin (ng) | 385.4±243.2 |
| KC/CXCL1 (pg) | 113350.4±7807.7 |
| Leptin (ng) | 134.2±69.1 |
| MCP-1/CCL2 (ng) | 9.6±10.2 |
| IP-10/CXCL10 (pg) | 8442.0±12772.5 |
| PAI-1 (pg) | 4861.3±1978.3 |
| IL-10 (pg) | 490.8±349.8 |
| MIP-1α/CCL3 (pg) | 102.5±64.9 |
| TNF-α (pg) | 117.2±103.5 |
| RANTES/CCL5 (pg) | 113350.4±78107.7 |
| IL-17 (pg) | 70.0±23.2 |

Concentrations of adipokines and inflammatory factors measured in the culture media after 4 hours of *ex-vivo* culture of the eWAT being removed at surgery (t=8 weeks). Data are shown as mean±SD. Abbreviations: (IL-10) interleukin 10; (IL-17) interleukin 17; (IP-10/CXCL10) interferon ɣ-inducible protein 10; (KC/CXCL1) keratinocyte chemoattractant; (MCP-1/CCL2) monocyte chemoattractant protein 1; (MIP-1α/CCL3) macrophage inflammatory protein 1α; (PAI-1) plasminogen activator inhibitor 1; (RANTES/CCL5) regulated on activation, normal T-cell expressed and secreted; (TNF-α) tumor necrosis factor α.

**Table 2. Proteins secreted within 4 hours by eWAT depots at the end of the study.**

| Proteins | Secretion capacity of eWAT depot | | |
|---|---|---|---|
| | Chow | HFD+sham | HFD+WATx |
| **Adiponectin (ng)** | **2645.2±1166.6[a]** | **2731.4±909.3[a]** | **1097.1±302.4[b]** |
| **Resistin (ng)** | **228.7±82.7[a]** | **114.7±90.9[b]** | **61.3±21.9[b]** |
| KC/CXCL1 (pg) | 201962.5±69493.5 | 187017.8±137460.5 | 93592.6±74809.2 |
| **Leptin (ng)** | **63.5±36.2[a]** | **122.4±37.6[b]** | **53.6±13.0[a]** |
| **MCP-1/CCL2 (ng)** | **24.0±10.0[a]** | **9.5±8.0[b]** | **5.4±4.3[b]** |
| **IP-10/CXCL10 (pg)** | **10911.2±8364.7[a]** | **1908.4±2367.6[b]** | **683.4±513.1[b]** |
| **PAI-1 (pg)** | **5461.9±1293.3[a]** | **3883.7±1824.6[a]** | **1390.3±429.2[b]** |
| IL-10 (pg) | 1748.3±1178.0 | 1142.6±1122.7 | 1029.0±832.1 |
| MIP-1α/CCL3 (pg) | 276.0±148.5 | 1114.4±978.3 | 150.6±117.5 |
| TNF-α (pg) | 260.0±80.84 | 287.5±120.3 | 371.3±262.4 |
| RANTES/CCL5 (pg) | 271.3±139.1 | 325.7±168.9 | 215.2±108.6 |
| IL-17 (pg) | 35.7±15.6 | 68.2±39.4 | 27.5±18.1 |

Concentrations of adipokines and inflammatory factors measured in culture media after 4 hours of ex-vivo culture of the eWAT depots collected at the end of the study (t=28 weeks). Data are shown as mean±SD. Statistical differences are indicated in bold; groups with the same superscript letters are statistically comparable (p>0.05) and groups with different superscript letters are statistically different (p≤0.05). Abbreviations: (IL-10) interleukin 10; (IL-17) interleukin 17; (IP-10/CXCL10) interferon ɣ-inducible protein 10; (KC/CXCL1) keratinocyte chemoattractant; (MCP-1/CCL2) monocyte chemoattractant protein 1; (MIP-1α/CCL3) macrophage inflammatory protein 1α; (PAI-1) plasminogen activator inhibitor 1; (RANTES/CCL5) regulated on activation, normal T-cell expressed and secreted; (TNF-α) tumor necrosis factor α.

and in case of sWAT, secretion capacities for adiponectin, IP-10/CXCL10 and PAI-1 were slightly lower in numerical terms, but these differences were not statistically significant (S3 Table in S3 File).

## 4. Effect of partial eWAT lipectomy on bioactive lipids and cytokines in plasma

We first assessed whether eWAT lipectomy had effects on bioactive mediators in the circulation, including plasma FFAs, cytokines and chemokines. FFA concentrations were measured in plasma 1 month after surgery by lipidomics (Table 3). Concentrations of several FFAs including saturated fatty acids (e.g., lauric acid, margaric acid, stearic acid), monounsaturated fatty acids (e.g., oleic acid) and polyunsaturated fatty acids (PUFAs, e.g., eicosadienoic acid, eicosatrienoic acid, arachidonic acid, adrenic acid, docosapentaenoic acid) were increased upon HFD feeding relative to Chow. HFD feeding also triggered decreases in some PUFAs including α-linolenic acid, linoleic acid, stearidonic acid and docosadienoic acid.

**Table 3. Concentrations of plasma FFAs measured at t = 14 weeks (1 month after surgery).**

| Plasma FFAs (nmol/ml) | Chow | HFD+sham | HFD+WATx |
|---|---|---|---|
| *Total SFAs* | *210.44* | *230.30* | *238.76* |
| **Lauric acid (12:0)** | **0.69[a]** | **0.82[b]** | **0.84[b]** |
| Myristic acid (14:0) | 8.14 | 8.58 | 8.44 |
| Pentadecanoic acid (15:0) | 4.83 | 6.29 | 5.30 |
| Palmitic acid (16:0) | 131.98 | 137.08 | 143.55 |
| **Margaric acid (17:0)** | **4.30[a]** | **5.25[b]** | **4.86[b]** |
| **Stearic acid (18:0)** | **54.09[a]** | **65.16[b]** | **68.68[b]** |
| Arachidic acid (20:0) | 2.81 | 2.90 | 2.98 |
| Behenic acid (22:0) | 2.24 | 2.56 | 2.58 |
| Lignoceric acid (24:0) | 1.36 | 1.66 | 1.52 |
| *Total MUFAs* | *174.78* | *194.94* | *210.25* |
| Myristoleic acid (14:1) | 2.54 | 3.09 | 2.90 |
| Palmitoleic acid (16:1) | 45.58 | 35.80 | 34.87 |
| **Oleic acid (18:1)** | **112.40[a]** | **142.37[b]** | **157.65[b]** |
| Eicosenoic acid (20:1) | 5.26 | 5.46 | 6.26 |
| Erucic acid (22:1) | 6.37 | 5.49 | 5.71 |
| Nervonic acid (24:1) | 2.63 | 2.74 | 2.86 |
| *Total PUFAs* | *163.15* | *145.92* | *164.79* |
| **Linoleic acid (18:2)** | **94.16[a]** | **64.27[b]** | **70.92[a,b]** |
| **α-linolenic acid (ALA, 18:3)** | **8.89[a]** | **5.76[b]** | **5.90[b]** |
| **Stearidonic acid (18:4)** | **0.52[a]** | **0.43[b]** | **0.43[b]** |
| **Eicosadienoic acid (20:2)** | **2.67[a]** | **3.75[b]** | **4.01[b]** |
| **Eicosatrienoic acid (20:3)** | **5.73[a]** | **7.61[b]** | **8.90[c]** |
| **Arachidonic acid (20:4)** | **24.66[a]** | **36.23[b]** | **43.77[b]** |
| Eicosapentaenoic acid (EPA, 20:5) | 4.84 | 4.00 | 4.22 |
| **Docosadienoic acid (22:2)** | **2.17[a]** | **1.66[b]** | **1.58[b]** |
| **Adrenic acid (22:4)** | **1.74[a]** | **2.68[b]** | **2.76[b]** |
| **Docosapentaenoic acid (DPA, 22:5)** | **4.54[a]** | **5.84[b]** | **5.87[b]** |
| **Docosahexaenoic acid (DHA, 22:6)** | **13.23[a]** | **13.68[a]** | **16.42[b]** |

Free fatty acid (FFA) concentrations were measured at t = 14 weeks (1 month after surgery). Data are shown in nmol/mL as mean ± SD. Statistical differences are indicated in bold; groups with the same superscript letters are statistically comparable (p > 0.05) and groups with different superscript letters are statistically different (p ≤ 0.05). Abbreviations: (MUFAs) monounsaturated fatty acids, (PUFAs) polyunsaturated fatty acids, (SFAs) saturated fatty acids.

Interestingly, when compared to HFD+sham, eWAT lipectomy specifically increased the PUFAs eicosatrienoic acid and docosahexaenoic acid (DHA), i.e., fatty acids with beneficial anti-inflammatory properties.

In addition, adipokines, cytokines, chemokines and metabolic factors were analyzed in plasma 4 months after surgery by ELISA and targeted proteomics (Olink®). ELISA analyses showed that HFD feeding increased plasma concentrations of leptin, SAA, S100B, MIF, PAI-1, IFN-γ, IL-10, KC/CXCL1, TNF-α, IP-10/CXCL10 and MCP-1/CCL2, none of which being significantly affected by the partial eWAT lipectomy (S4 Table in S3 File).

Proteomics further revealed elevated levels of several pro-inflammatory factors (e.g., MCP-1/CCL2, MIP-1β/CCL4, IL-1α, IL-3, IL-6, TNF), but also metabolic factors (e.g., fibroblast growth factor 2 (FGF21) and some anti-inflammatory factors (e.g., IL-10) in HFD+sham mice compared with Chow mice (Table 4). Notably, IL-22 levels were specifically decreased in HFD+sham mice. Furthermore, in HFD + WATx mice, IFN-γ levels were reduced while IL-17F concentrations were elevated compared with Chow mice. However, no significant differences in protein concentrations were observed between HFD + WATx and HFD+sham mice. Of note, eWAT lipectomy seemed to impact several cytokines and chemokines in a counterregulatory way based on numerical trends of their absolute concentrations compared with HFD+sham. Absolute concentrations (in ng/mL) of each protein are provided in S5 Table in S3 File.

In parallel, multiplex-based quantification of cytokines (IL-10, IL-15, IL-1β, IL-33, IL-6, TNF-α) and other markers (BDNF, MIP-1α/CCL3) in brain cortex homogenates showed no significant differences between the three groups (S6 Table in S3 File).

## 5. Partial eWAT lipectomy does not affect brain structure

Cortical thickness and hippocampal volume were measured in T2-weighted anatomical images (S3A-B Fig in S2 File). At=7 weeks of HFD feeding (prior to surgery), the HFD group showed slightly larger cortical thickness than the Chow mice (HFD $0.86 \pm 0.03$ vs Chow $0.85 \pm 0.02$, p=0.049). After lipectomy, no differences in cortical thickness were observed between the groups. Hippocampal volume remained the same over time for the HFD-sham, HFD + WATx and Chow groups (p=0.222). Grey and white matter integrity were further assessed with DTI (S3C-F Fig in S2 File). In the Chow group, fractional anisotropy increased over time between t=12 weeks and t=27 weeks, both in white (p=0.004) and grey matter (p=0.017). Notably, at t=27 weeks, the HFD + WATx mice showed significantly lower fractional anisotropy in grey matter than Chow mice (p=0.029). Mean diffusivity in white and grey matter remained constant over time and did not differ between the groups.

## 6. Partial eWAT lipectomy reverses HFD-induced increase in CBF under vasoconstrictive conditions

**6.1 CBF under normal gas mix.** CBF was first measured under normal gas mix (1:2 oxygen – medical air, Fig 3A-D). At t=7 weeks of HFD feeding (prior to surgery), no differences in CBF were observed between Chow and HFD groups. After surgery, the HFD+sham group showed increased CBF compared with the Chow group (overall effects: cortex p=0.007, hippocampus p=0.010 and thalamus p=0.029). The CBF measured in HFD + WATx mice seemed somewhat lower than in HFD+sham mice (−9% on average) reverting towards Chow, but there were no statistical differences between the HFD + WATx and HFD+sham groups (overall effects: cortex p=0.276, hippocampus p=0.485, thalamus p=0.233) or between the HFD + WATx and Chow groups (overall effects: cortex p=0.313, hippocampus p=0.068, thalamus p=0.672).

CBF outcomes measured under normal gas mix were additionally analyzed over time between pre-surgery (t=7 weeks) and post-surgery time points (t=12 weeks and t=26 weeks) in Chow, HFD+sham and HFD + WATx groups, to further characterize the effect of aging on CBF. For this specific analysis at t=7 weeks (prior to surgery), the combined HFD group was retrospectively divided into HFD+sham and HFD + WATx subgroups, mirroring the group separation used at the post-surgery time points. No time effect was observed in the cortex (overall time effect: p=0.099). In the hippocampus, CBF significantly decreased over time in all the groups (overall time effect: p=0.012). In the thalamus, CBF also

**Table 4. Multiplex analysis of plasma proteins at t = 27 weeks (4 months after surgery).**

| Protein | Normalized Protein Expression (NPX) units | | |
|---|---|---|---|
| | Chow | HFD+sham | HFD+WATx |
| Eotaxin-1/CCL11 | 10.78±0.44 | 11.22±0.35 | 11.09±0.36 |
| **MCP-5/CCL12** | **7.98±0.38[a]** | **8.65±0.36[b]** | **8.67±0.52[b]** |
| TARC/CCL17 | 2.76±1.09 | 3.47±0.98 | 2.67±0.92 |
| **MCP-1/CCL2** | **10.76±0.58[a]** | **12.16±0.54[b]** | **12.05±0.61[b]** |
| MDC/CCL22 | 12.33±0.55 | 12.60±0.53 | 12.45±0.57 |
| **MIP-1β/CCL4** | **7.53±0.31[a]** | **8.65±0.37[b]** | **8.76±0.57[b]** |
| RANTES/CCL5 | 5.63±0.65 | 6.15±0.75 | 5.79±0.47 |
| **PD-L1/CD274** | **4.21±0.34[a]** | **4.71±0.31[b]** | **4.64±0.24[b]** |
| M-CSF/CSF1 | 7.71±0.33 | 7.82±0.34 | 7.91±0.22 |
| GM-CSF/CSF2 | 1.58±0.24 | 1.75±0.34 | 1.70±0.43 |
| G-CSF/CSF3 | 7.30±0.66 | 7.49±0.51 | 7.78±0.55 |
| CTLA-4 | 3.93±0.29 | 4.19±0.36 | 4.10±0.21 |
| **KC/CXCL1** | **11.77±0.36[a]** | **12.67±0.39[b]** | **12.71±0.5[b]** |
| I-TAC/CXCL11 | 0.44±0.27 | 0.65±0.3 | 0.77±0.24 |
| **MIP-2/CXCL2** | **6.15±0.58[a]** | **7.62±0.58[b]** | **7.56±0.71[b]** |
| **MIG/CXCL9** | **9.39±0.51[a]** | **10.53±0.51[b]** | **10.34±0.61[b]** |
| **FGF21** | **10.58±1.36[a]** | **12.70±1.04[b]** | **12.45±1.31[b]** |
| HGF | 9.41±3.64 | 8.51±4.69 | 7.46±5.07 |
| **IFN-α2** | **3.14±0.46[a]** | **3.9±0.39[b]** | **4.18±0.32[b]** |
| **IFN-γ** | **4.24±0.51[a]** | **3.86±0.34[a,b]** | **3.65±0.35[b]** |
| **IFN-λ2** | **1.82±0.23[a]** | **2.21±0.58[b]** | **2.25±0.41[b]** |
| **IL-10** | **3.38±0.28[a]** | **4.39±0.34[b]** | **4.49±0.53[b]** |
| IL-12α/IL-12β | 2.66±1.28 | 1.44±0.64 | 1.62±0.9 |
| IL-16 | 8.60±0.27 | 8.84±0.36 | 8.86±0.43 |
| IL-17A | 5.30±1.03 | 5.70±0.61 | 6.05±0.75 |
| **IL-17F** | **6.17±1.35[a]** | **7.47±1.18[a,b]** | **7.92±0.94[b]** |
| **IL-1α** | **8.44±1.13[a]** | **10.29±1.08[b]** | **9.83±1.59[b]** |
| IL-1β | 2.86±0.59 | 3.23±0.67 | 3.15±0.55 |
| IL-2 | 2.93±0.27 | 3.15±0.34 | 3.06±0.28 |
| IL-21 | 0.07±0.38 | −0.02±0.2 | 0.28±0.27 |
| **IL-22** | **4.83±0.72[a]** | **4.05±0.98[b]** | **3.64±0.49[b]** |
| **IL-27** | **2.35±0.43[a]** | **3.36±0.49[b]** | **3.35±0.52[b]** |
| **IL-3** | **0.50±0.48[a]** | **0.80±0.46[a,b]** | **0.99±0.26[b]** |
| IL-33 | 2.16±0.87 | 2.83±0.54 | 2.32±0.75 |
| IL-4 | 1.21±0.27 | 1.40±0.41 | 1.39±0.3 |
| IL-5 | 4.14±0.55 | 4.62±0.85 | 4.34±0.33 |
| **IL-6** | **2.71±1.12[a]** | **4.21±1.1[b]** | **4.48±0.96[b]** |
| IL-7 | 1.38±0.45 | 1.52±0.49 | 1.67±0.36 |
| IL-9 | 4.83±0.96 | 5.36±1.47 | 4.43±0.48 |
| PD-L2/PDCD1LG2 | 13.15±0.30 | 13.21±0.28 | 13.24±0.23 |
| **TNF** | **3.45±0.54[a]** | **4.65±0.56[b]** | **4.66±0.58[b]** |

Multiplexed quantification of proteins was performed in plasma samples collected t = 27 weeks. Protein levels are expressed in Normalized Protein expression (NPX) units on a log2 scale. Data are shown as mean ± SD. Statistical differences are indicated in bold; groups with the same superscript letters are statistically comparable (p > 0.05) and groups with different superscript letters are statistically different (p ≤ 0.05). Abbreviations: (CTLA-4) cytotoxic T-lymphocyte associated protein 4, (FGF21) fibroblast growth factor 2, (G-CSF/CSF3) granulocyte colony-stimulating factor, (GM-CSF/CSF2)

*(Continued)*

**Table 4.** (Continued)

granulocyte-macrophage colony-stimulating factor, (HGF) hepatocyte growth factor, (I-TAC/CXCL11) interferon-inducible T-cell alpha chemoattractant, (IFN) interferon, (IL) interleukin, (KC/CXCL1) keratinocyte-derived chemokine, (M-CSF/CSF1) macrophage colony-stimulating factor, (MCP-1/CCL2) monocyte chemoattractant protein 1, (MCP-5/CCL12) monocyte chemoattractant protein 5, (MDC/CCL22) macrophage-derived chemokine, (MIG/CXCL9) monokine induced by gamma interferon, (MIP-1β/CCL4) macrophage inflammatory protein 1β, (MIP-2/CXCL2) macrophage inflammatory protein 2, (PD-L1/CD274) programmed death-ligand 1, (PD-L2/PDCD1LG2) programmed death-ligand 2, (RANTES/CCL5) regulated on activation, normal T-cell expressed and secreted, (TARC/CCL17) thymus and activation-regulated chemokine, (TNF) tumor necrosis factor.

decreased over time in the Chow group (time effect: p = 0.002) and the HFD + WATx group (time effect: p = 0.001), while it remained stable in the HFD+sham group (time effect: p = 0.715).

### 6.2 CBF under vasoconstrictive conditions (induced by pure oxygen)

Next, the mice were challenged with pure oxygen to induce cerebral vasoconstriction and thereby assess the responsivity of the vasculature in a functional test. At t = 7 weeks of HFD feeding, no differences between Chow and HFD groups were observed. After surgery however, CBF during vasoconstriction was higher in the cortex and thalamus of HFD+sham mice than in Chow mice (overall effects: cortex p = 0.049 and thalamus p = 0.039, Fig 3E,G). In the hippocampus at t = 12 weeks, CBF was also higher in the HFD+sham and HFD + WATx mice than in Chow mice (HFD+sham vs Chow p = 0.037, HFD + WATx vs Chow p = 0.037, Fig 3F). Thereafter, between t = 12 weeks and t = 27 weeks, hippocampal CBF tended to decrease over time in the HFD + WATx group (p = 0.076 within group) while it remained constant in the Chow and HFD+sham groups. This gradual decrease in the HFD + WATx group ultimately resulted in a significantly lower hippocampal CBF compared with the HFD+sham group (HFD + WATx vs HFD+sham p = 0.015), which then was comparable to the Chow reference (HFD + WATx vs Chow p = 0.828). This result indicates an increased cerebral vasoreactivity and improved vascular function in the hippocampus as a consequence of eWAT lipectomy (Fig 3I). Of note, cerebral vasoreactivity in the cortex and thalamus were not altered by eWAT lipectomy (Fig 3H, J).

### 7. Partial eWAT lipectomy increases rs-FC connectivity between the hippocampus and cortical regions

The resting state functional connectivity (rs-FC) outcomes based on total correlation analyses are provided in an overview matrix in Fig 4A,C. At t = 7 weeks (prior to surgery), the interregional rs-FC with somatosensory cortex was lower in the HFD group compared with Chow group (Fig 4B). After surgery, a global increase in connectivity between multiple regions of the brains was observed over time in all groups (Fig 4D) and differences between the HFD+sham group and the Chow controls were no longer observed (Fig 4E). Compared with the Chow group, HFD + WATx mice showed increases in rs-FC between the left ventral hippocampus and right dorsal hippocampus, auditory, motor and visual cortices, as well as between the dorsal hippocampus and motor cortex within the right hemisphere. However, no differences between the HFD + WATx and HFD+sham groups or between the HFD+sham and Chow groups were observed.

### 8. Partial eWAT lipectomy does not affect spatial learning and memory performance

At t = 6 weeks (prior to surgery), no difference in learning abilities or memory performance were observed between Chow and HFD groups during the MWM test (S4 Fig in S2 File). At t = 11 weeks (i.e., 1 month after surgery), an ORT was performed over 3 days to assess general behavior and short-term memory but no differences were observed between Chow, HFD+sham and HFD + WATx groups (Fig 5A-C). At t = 26 weeks, a reverse MWM test was performed to assess the long-term effect of HFD feeding and partial eWAT lipectomy on spatial learning and memory. During the first probe test (assessment of long-term memory), no difference was observed between the groups (all parameters p > 0.05, Fig 5D-E). Next, over the 2 days of the new acquisition phase (learning phase for a new platform location), only the Chow mice showed a reduction in latency and distance moved to find the platform over time (p = 0.008 and p = 0.004 respectively,

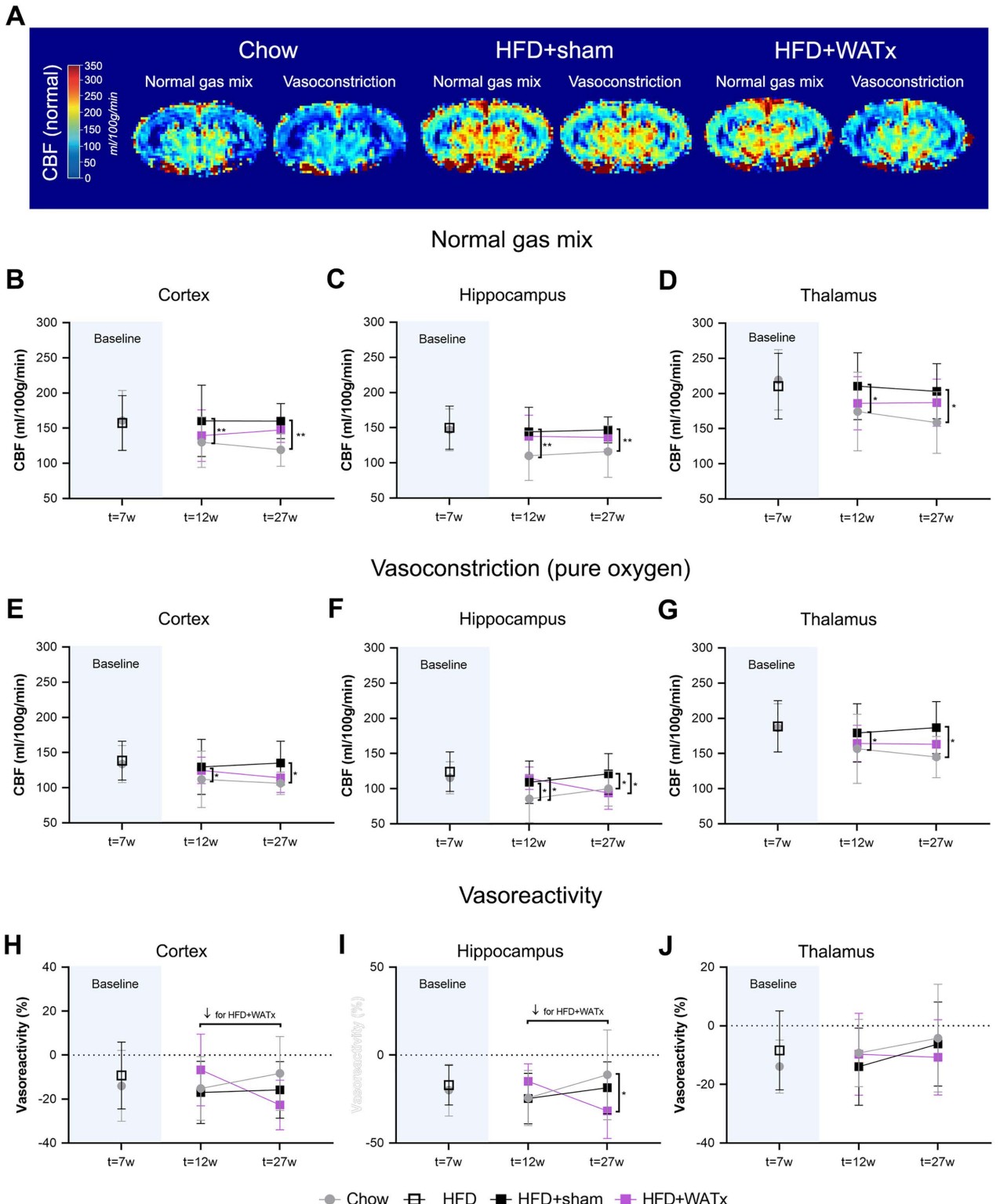

**Fig 3. Brain blood perfusion and cerebral vasoreactivity.** (A) Representative pictures of cerebral blood flow (CBF) measured with arterial spin labeling under normal gas mix and under pure oxygen (vasoconstriction) in Chow, HFD+sham and HFD+WATx mice at t=27 weeks (i.e., 4 months after the surgery). CBF was measured under normal gas mix in the (B) cortex, (C) hippocampus, (D) thalamus at t=7 weeks (prior to surgery), t=12 weeks

(1 month after surgery), and t = 27 weeks (4 months after surgery). (E-G) CBF was also measured under pure oxygen (vasoconstrictive conditions) in the same brain regions and at the same time points. For each time point, (H-J) cerebral vasoreactivity was calculated as the difference between CBF in vasoconstrictive conditions and CBF in normal conditions normalized by the CBF in normal conditions. Data are shown as mean ± SD. # p ≤ 0.05 over time. * p ≤ 0.05, ** p ≤ 0.01, *** p ≤ 0.001 between the experimental groups.

Fig 5F-G). Of note, the average velocity remained the same between the three groups (p = 0.418, not shown). During the second probe test (assessment of short-term memory), the HFD + WATx mice spent significantly less time in the platform zone than the HFD+sham mice (p = 0.002, Fig 5H), and both Chow and HFD + WATx groups crossed the platform zone less often than the HFD+sham group (HFD+sham vs Chow p = 0.039, HFD + WATx vs HFD+sham p < 0.001, HFD + WATx vs Chow p = 0.150, Fig 5I).

## 9. Partial eWAT lipectomy alleviates HFD-induced sedentary behavior

Home-cage (DVC) activity was monitored during day time and night time throughout the study (Fig 6). Within the light and dark phase, activity decreased over time for all the groups as mice became older. No group differences were observed until 25 weeks of HFD feeding. After 25 weeks of HFD feeding, HFD+sham mice showed a more sedentary phenotype during the night (the period in which mice are typically active) and their night activity was comparably low as their day activity. By contrast, Chow and HFD + WATx mice did not show this sedentary phenotype and showed an increased activity in the night time compared with the day time throughout the entire study, a behavioral pattern which is typical for mice.

## 10. Transcriptomics analysis predicts a specific inactivation of USP22 in the hippocampus upon eWAT lipectomy

To gain insight into the effects of HFD feeding and partial eWAT lipectomy on brain pathology, the brains were analyzed immunohistopathologically at the end of the study (S5 Fig in S2 File). No group differences were observed in markers of microglia activation (IBA-1-positive area), astrogliosis (GFAP-positive area and GFAP staining intensity), cerebrovascular integrity (GLUT-1-positive area), and neurogenesis (DCX-positive neurons). In parallel, multiplex-based quantification of cytokines (IL-10, IL-15, IL-1β, IL-33, IL-6, TNF-α) and other markers (BDNF, MIP-1α/CCL3) in brain cortex homogenates showed no significant differences between the three groups (S6 Table in S3 File).

To examine potential biological processes that may underlie the effects of HFD feeding and partial eWAT lipectomy on the brain, next generation RNA-sequencing was performed on hippocampi, to define differential activation of canonical pathways and upstream regulators that control critical biological processes. Compared with the Chow group, HFD feeding significantly activated pathways related to hippocampal inflammation. Among these pathways were "Neuroinflammation signaling" (-log(p)=3, z-score = 2.4), "Macrophage classical activation signaling" (-log(p)=2.9, z-score = 2.4) and the "Complement system" pathway (trend activation (-log(p)=8.5, z-score = 1.3) as detailed in S6A Fig in S2 File and S7 Table in S3 File. Consistent with this, upstream regulators implicated in inflammatory responses were also activated by HFD feeding (e.g., tumor-necrosis factor: -log(p)=6.2, z-score = 3.3; interferon-γ: -log(p)=8.9, z-score = 4.1), as detailed in S6B Fig in S2 File and S8 Table in S3 File. In addition, several upstream regulators predicted to be upregulated in the HFD+sham group are associated with the type I interferon (IFN)/JAK/STAT signaling pathway, including signal transducer and activator of transcription 1 (STAT1), interferon α/β receptor (IFNAR), interferon α2 (IFNA2), interferon regulatory factor 3 (IRF3), and interferon regulatory factor 7 (IRF7).

Canonical pathways did not differ significantly between HFD + WATx and HFD+sham groups (all -log(p)<2), but on the level of upstream regulators, the enzyme ubiquitin specific peptidase 22 (USP22) was predicted to be significantly inactivated upon lipectomy in HFD + WATx mice compared with HFD+sham mice (-log(p)=3.6, z-score = −2.0) as detailed in S6C Fig in S2 File and S9 Table in S3 File. The role of USP22 in detrimental HFD-induced hippocampal changes is further supported by the observation that USP22 activation was predicted to be higher (trend) in HFD+sham compared with Chow (HFD+sham vs Chow mice (-log(p)=9.2, z-score = 1.0).

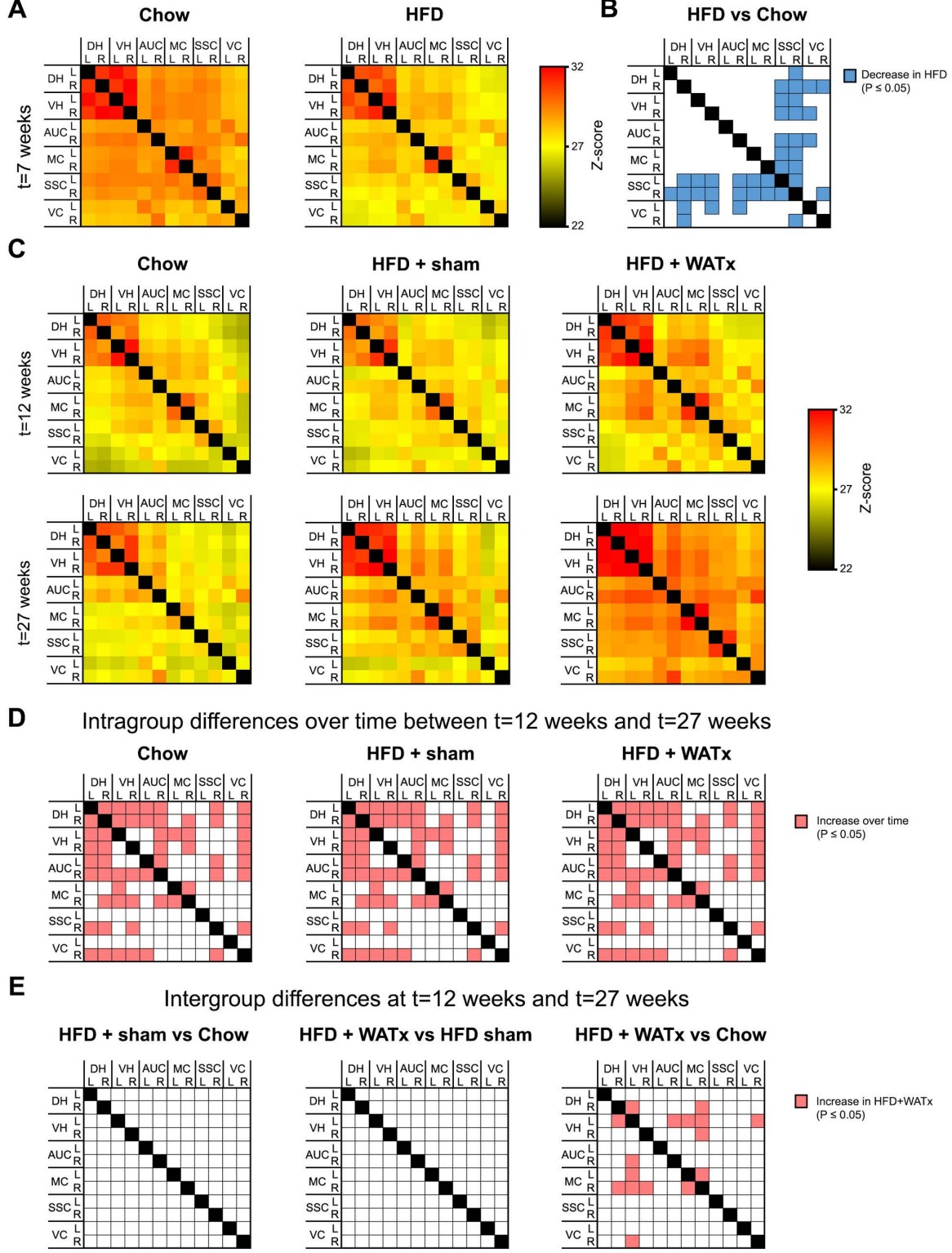

**Fig 4. Resting-state functional connectivity (rs-FC).** Rs-FC between brain regions was measured with resting-state fMRI. (A) Total correlation matrixes showing the rs-RC between the brain regions calculated at t = 7 weeks (prior to surgery). (B) Statistical differences in rs-FC between HFD and Chow groups at t = 7 weeks. (C) Total correlation matrixes showing the rs-RC between the brain regions calculated at t = 12 weeks (1 month after

surgery), and t = 27 weeks (4 months after surgery). (D) Statistical changes in rs-FC within Chow, HFD+sham and HFD + WATx groups over time between t = 12 weeks and t = 27 weeks. (E) Statistical differences in rs-FC between Chow, HFD+sham and HFD + WATx groups at t = 12 weeks and t = 27 weeks. A higher z-score indicates higher rs-FC. Abbreviations: (DH) dorsal hippocampus; (VH) ventral hippocampus; (AUC) auditory cortex; (MC) motor cortex; (SSC) somatosensory cortex; (VC) visual cortex in left (L) and right (R) hemispheres.

## Discussion

In the present study, we investigated the impact of partial (~70%) visceral eWAT lipectomy on brain health and obesity-associated comorbidities. The surgically removed fat pad was demonstrated to release a plethora of adipokines and inflammatory mediators (e.g., adiponectin, resistin, leptin, IP-10, KC, TNF-α, RANTES) in *ex vivo* culturing experiments. Histological examination at the end of the study revealed that partial eWAT lipectomy reduced the number of severely hypertrophic adipocytes of the remaining residual-eWAT, and altered its properties to secrete adipokines and inflammatory mediators *ex vivo*. The partial lipectomy increased specific circulating FFAs that have anti-inflammatory and vaso-protective properties (e.g., DHA, eicosatrienoic acid), but also attenuated some detrimental effects of HFD-induced obesity on the brain itself: It prevented HFD-induced impairment of cerebral vasoreactivity in the hippocampus and ameliorated cortico-hippocampal connectivity unrelated to body weight or food intake. In addition, partial lipectomy alleviated the development of HFD-induced sedentary behavior (i.e., reduced night time activity) and partly altered the performance in a spatial memory task.

Partial eWAT lipectomy was performed after 8 weeks of HFD feeding, i.e., when the eWAT depot is maximally expanded but not yet inflamed in HFD-treated Ldlr-/-.Leiden mice based on previous longitudinal studies [22]. As expected, the eWAT pads at the end of the study (t = 28 weeks) were significantly smaller in the mice that underwent partial eWAT lipectomy than in sham-operated mice or Chow control mice. Furthermore, the weights of the other WAT depots (mWAT and sWAT) and CLS-based inflammation in all WAT depots were similar between the lipectomized mice and HFD-fed sham controls, indicating that the other WAT depots did not compensate for lipectomy with additional fat storage or inflammation. Interestingly, the partial eWAT lipectomy reduced the proportion of very large adipocytes (>8000 μm²) in the residual-eWAT. Skurk *et al.* notably reported that specifically the very large adipocytes are dysregulated and prone to release pro-inflammatory mediators [19]. In the present study, the reduction of very large adipocytes in the residual-eWAT of lipectomized mice was accompanied by changes in eWAT secretome as demonstrated by 4-hour *ex vivo* culture experiments at the end of the study: Leptin and adiponectin secretion capacities were lower compared with HFD-fed sham controls, an observation that is consistent with reported positive associations between secretion of leptin and adiponectin and adipocyte volume [19]. In addition, after partial eWAT lipectomy, the secretion of PAI-1 by the residual-eWAT was also significantly reduced. PAI-1 is a major inhibitor of the fibrinolytic system which is notably involved in atherosclerosis, thrombosis and metabolic dysfunction [37] and, interestingly, the inhibition of PAI-1 has been shown to improve cerebral perfusion in ischemic stroke models [38]. In addition to PAI-1, the secretion rates of MIP-1α/CCL3 and IL-17 by the residual-eWAT were numerically lower (about −86% and −60% respectively) compared with HFD+sham controls. Although effects did not reach statistical significance these results suggest that partial lipectomy can affect the release of pro-inflammatory factors from the remaining residual-eWAT. Notably, both MIP-1α/CCL3 and IL-17 have been associated with impaired synaptic plasticity and poorer memory function [39,40], suggesting that eWAT lipectomy partly prevented the release of proteins associated with detrimental effects on the brain.

eWAT lipectomy had no significant effect on inflammatory plasma proteins: Multiple inflammatory markers were increased in HFD+sham mice compared with Chow controls but there were no statistically significant differences between HFD+sham and HFD + WATx mice, albeit several cytokines and chemokines seemed to be impacted in a counterregulatory way based on the absolute concentrations. This may contradict results from others showing significant reductions in TNF-α and IL-1β levels in the circulation of eWAT-lipectomized mice [41]. This may be due to

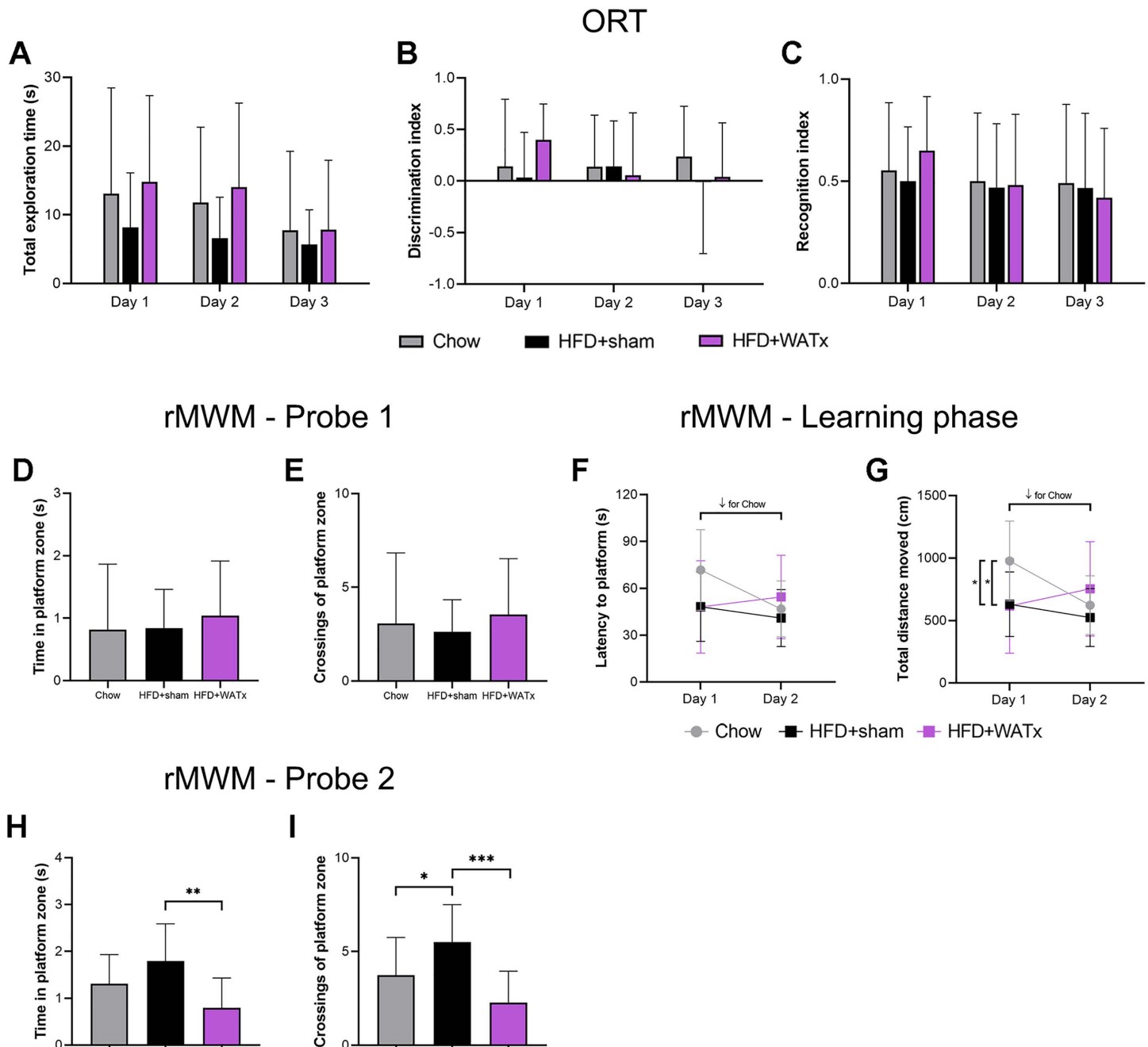

**Fig 5. Spatial learning, memory performance and explorative behavior after surgery.** An Object Recognition Test (ORT) was performed at t = 11 weeks (1 month after the surgery) to asses short-term memory and explorative behavior: (A) total exploration time, (B) discrimination index, and (C) recognition index over the 3 days of ORT. A reverse Morris Water Maze (rMWM) was performed at t = 26 weeks (4 months after the surgery) to assess (D-E) long-term memory (Probe 1), (F-G) spatial learning (learning phase, 2 days), and (H-I) short-term memory (Probe 2). Data are shown as mean ± SD. ↓ decrease over time for Chow group (p ≤ 0.05). * p ≤ 0.05, ** p ≤ 0.01, *** p ≤ 0.001 between the experimental groups.

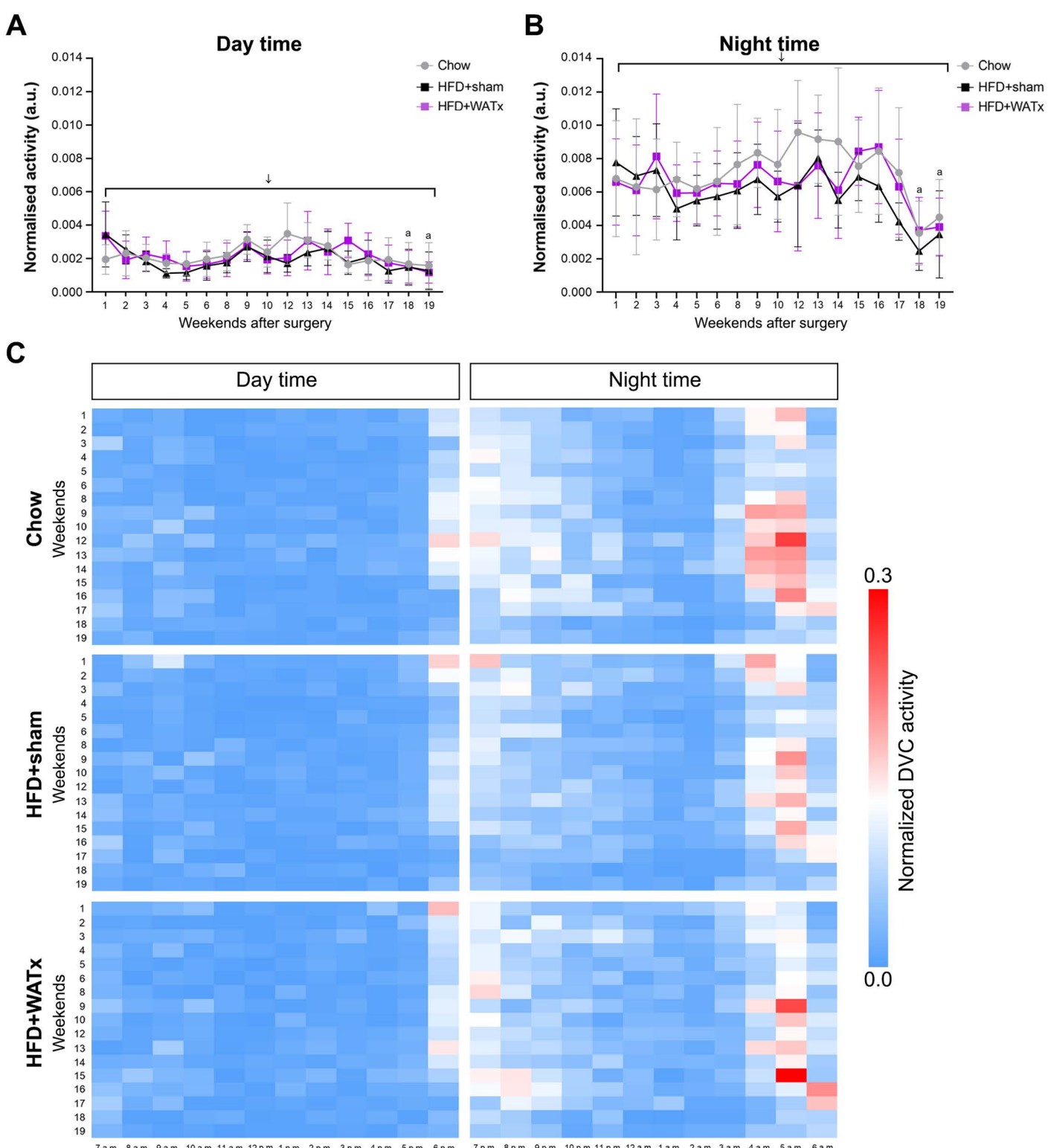

**Fig 6. Home-cage activity.** DVC data were analyzed in the weekends after surgery (from t = 9 week onwards) until the end of the study (t = 28 weeks). Average weekend activity during (A) day time and (B) night time. (C) Heat-map representing average activity per hour during the day time and night time. ↓ decrease over time for all the groups (p ≤ 0.05). [a] In HFD+sham group night activity was similar to day activity from weekend 18.

a difference in experimental conditions since lipectomy was performed on a chow diet (i.e., leaner mice), while the present study was performed in the context of a much more severe condition, HFD-induced obesity. Under the conditions employed herein many other metabolically dysfunctional and inflamed tissues (e.g., liver, vasculature, pancreas) as well as immune cells can contribute to the plasma pool of inflammatory factors, and potentially mask the effects of eWAT lipectomy. This may also explain why we did not observe significant effects on metabolic profile (e.g., body weight, plasma cholesterol and triglyceride levels) and metabolic comorbidities (e.g., liver disease, atherosclerosis). Since eWAT lipectomy in this study was only partial, the difference in secretory proteins released into the circulation might be therefore too small to detect with current techniques. Of note, although effects did not reach statistical significance regarding a specific cytokine or chemokine, it is still possible that the net effect of the many insignificant reductions of cytokines and chemokines together may impact the brain. Moreover, the sample size may have been insufficient to detect statistically significant differences. Increasing the sample size in future studies could help better characterize the impact of eWAT lipectomy on circulating inflammatory markers. Future studies should also consider to select specific cytokines and chemokines on basis of our findings allowing a more targeted (statistical) analysis, and to measure fluxes of WAT-derived cytokines with kinetic labeling, rather than their steady-state concentrations, for a more comprehensive understanding of systemic processes.

While eWAT lipectomy did not significantly affect cytokine concentrations in plasma, it did however modulate the levels of specific circulating FFAs. Following lipectomy, the levels of the PUFAs eicosatrienoic acid and DHA increased, both of which have been shown to possess anti-inflammatory properties [42–44]. Moreover, higher DHA levels have been suggested to have beneficial effects on brain blood perfusion and blood-brain barrier integrity [45,46]. Altogether, the observed elevations of FFAs with anti-inflammatory and vaso-protective properties may provide a mechanistic rationale for the observed effects of eWAT lipectomy on the brain. We cannot exclude that cytokines or chemokines are also mediating such effects because we employed an untargeted (48 panel) analysis which requires correction for multiple testing. Future lipectomy studies should include cytokine/chemokine measurements, because the numerical changes of plasma cytokine and chemokine concentrations suggest that lipectomy may counter-regulate the detrimental effects of HFD feeding, although statistical significance was not reached in the present study. Future studies evaluating the effect of WAT lipectomy could use our dataset to select cytokines/chemokines for more targeted analyses.

In the present study, prolonged HFD feeding (>12 weeks) was associated with a higher CBF in comparison with mice fed a chow diet. In the thalamus especially, this HFD-induced increase in CBF seems to be due to a time-related decline in CBF in chow-fed mice, which is not observed in the HFD-fed mice. We therefore hypothesize that, under chow conditions, the process of aging tends to decrease CBF but, in obese (HFD) conditions, this decline is masked by a pathological increase of CBF that is associated with impaired vascular function of smaller vessels (e.g., capillaries) and larger vessels (e.g., atherosclerosis in aortic root). A higher CBF upon HFD feeding may seem unexpected, as previous studies in HFD-fed Ldlr-/-.Leiden mice and obese patients often reported CBF decreases due to hypertension, atherosclerosis, impaired insulin-dependent vasodilation, oxidative stress-related endothelial injury, or pro-inflammatory cytokines [13,21,26]. However, consistent with our finding, other studies in individuals with obesity have described increases in CBF in certain regions of the brain [13]. For instance, Thomas *et al.* further found that the relationship between CBF and cognitive decline may be disease-stage dependent and bell-shaped, showing that patients with early-stage cognitive decline can have higher CBF in critical brain regions (e.g., hippocampus) compared to those with normal cognition or more advanced mild cognitive impairment. Hence, increases in cerebral perfusion may constitute a compensatory early-stage disease phenomenon in an attempt of the brain to maintain blood, nutrient and oxygen supply as an adaptive response to early neurodegenerative processes [47]. These findings would imply that the mice in the present study exhibit early stages of brain pathology which would also be in line with our observation that HFD feeding did not cause significant structural changes in the brain (e.g., hippocampus volume, cortical thickness) but already affected learning abilities as shown in the rMWM test.

Interestingly, an age-related decline in CBF similar to chow-fed mice was observed in the thalamus of HFD-fed eWAT-lipectomized mice, suggesting that the eWAT lipectomy may attenuate the HFD-induced increase in CBF. In addition, partial eWAT lipectomy lowered hippocampal CBF under vasoconstrictive conditions (induced by pure oxygen) towards levels of the Chow group, while CBF remained higher in HFD+sham control group. This decrease in CBF under vasoconstrictive conditions indicates an amelioration of cerebral vasoreactivity in the hippocampus, which is consistent with previous research in patients with obesity showing associations between visceral WAT dysfunction and disturbed brain perfusion [17,20]. Our results are also in line with studies reporting beneficial effects of eWAT lipectomy on age-induced BBB permeability and brain damage in the context of ischemic brain injury [41].

In the present study showed that HFD feeding can alter white matter microstructure: Based on DTI indices, we observed an increase in fractional anisotropy in Chow mice over time, but not in HFD-treated animals. This increase in fractional anisotropy in chow-fed animals specifically may reflect an improvement of white matter microstructure that may be attributed to a natural maturation of white matter tracts since myelination and neuronal growth have been shown to continue in young adult mice, leading to increased fractional anisotropy values [48,49]. The gradual increase in fractional anisotropy observed in the present study may, in part, also be ascribed to the repetition of learning tasks: Studies in humans and animals have indeed shown that cognitive and memory training, as well as learning tasks, can increase fractional anisotropy [50–53], suggesting an improvement of the connection between brain areas involved in cognitive tasks through an increase in nerve fibers or myelination. The lacking increase in fractional anisotropy in HFD-treated mice may suggest that HFD feeding could hamper the natural maturation of the white matter tract or impair the capacity to modulate white matter microstructure in response to learning. Consistent with this, in the rMWM test performed at the end of the study (t = 26 weeks), Chow mice successfully learned the new platform location, while this learning ability was impaired in mice on HFD (both eWAT-lipectomized mice and sham controls).

We additionally showed that HFD feeding and eWAT lipectomy induce distinct changes in brain functional connectivity. At t = 7 weeks of HFD feeding (prior to surgery), HFD-fed mice showed a reduction in interregional rs-FC between the somatosensory cortex and the hippocampus, and between somatosensory cortex and other cortical regions, compared with Chow controls. At the later time points, these differences were no longer observed suggesting that rs-FC may also be affected in Chow mice, for instance as a consequence of aging. Interestingly, one month after surgery, eWAT-lipectomized mice exhibited increased interhemispheric connectivity between the hippocampus and various cortical regions, which is vital for memory and cognition [54]. Previous studies in humans have shown that an improved navigational task performance is linked to increased connectivity between these regions [55]. In the present study, although partial eWAT lipectomy did not improve spatial learning abilities in the rMWM, eWAT-lipectomized mice showed similar alternative search patterns as the Chow mice in the probe test – i.e., both groups explored the platform zone less frequent than the HFD+sham mice when the platform was removed, suggesting a quicker realization of the platform's absence. The increase in cortico-hippocampal connectivity and the aforementioned improvement of hippocampal vasoreactivity may support a better performance, as these processes are associated with enhanced cognition [56] and better performance in episodic memory tasks [57]. However, this is a subtle observation and no clear effect of the eWAT lipectomy was observed on cognitive performance, which may be due to the fact that the effect of eWAT lipectomy on cortico-hippocampal connectivity and hippocampal vasoreactivity was very mild. Despite correlations between cerebral blood flow outcomes and cognitive performance in this mouse model [36], the effect of eWAT lipectomy may not be large enough to trigger significant changes in cognitive function. Furthermore, obesity effect on brain function is complex and multifactorial and other detrimental processes that were not affected by eWAT lipectomy in this study, such as insulin resistance, dyslipidemia (e.g., high LDL plasma levels), vascular dysfunction (e.g., atherosclerosis development), may still cause detrimental effects and hence maintain obesity-induced cognitive deficits [58–60].

Despite the absence of substantial effects on cognitive performance, the eWAT lipectomy affected important behavioral aspects: After 25 weeks of HFD feeding, HFD+sham mice exhibited reduced home-cage night activity towards

their day-time activity levels (the inactive period in mice), indicative of a more sedentary behavior. Interestingly, the partial eWAT lipectomy prevented the development of this HFD-induced sedentary behavior as eWAT-lipectomized mice remained active at night similarly to Chow reference mice. Future studies are however needed to unravel potential mechanisms linking visceral WAT dysfunction, locomotor activity and circadian rhythms.

The effects of HFD feeding and partial eWAT lipectomy on brain function were not accompanied by changes in brain immunohistopathology readouts (e.g., IBA-1, GFAP, GLUT1). We observed no changes in microglia activation and astrogliosis (indicative of neuroinflammation) between Chow and HFD-treated mice suggesting that either mice develop aging-related neuroinflammation already on chow diet, or that HFD feeding was not long or potent enough to trigger detectable neuroinflammation on histological level, or that neuroinflammation is reflected by changes in microglial markers other than IBA-1 (e.g., CD68). The addition of healthier (younger) groups in future studies may further help to delineate the effect of aging (on a chow maintenance diet) and HFD feeding regarding the development of brain (histo)pathology. To gain insight into the biological processes impacted by HFD-induced obesity and partial eWAT lipectomy that may not be detectable on histological level, RNA sequencing and subsequent pathway enrichment analyses were performed using hippocampi collected at sacrifice. In comparison with the Chow, HFD treatment activated multiple pathways and upstream regulators critical for neuroinflammation, but the partial eWAT lipectomy did not significantly alter any canonical pathway compared with HFD+sham mice. The upstream regulator analysis (i.e., analysis of key regulators that orchestrate and control biological processes based on the observed downstream gene transcripts) of hippocampal RNAseq data predicted a possible (yet specific) inactivation of USP22 upon eWAT lipectomy. USP22, a deubiquitylating enzyme involved in modifying misfolded proteins, has been shown to play an important role in cancer and neurological disorders by orchestrating the proteosomal degradation of dysfunctional (e.g., incorrectly folded) proteins [61], which are typically formed under conditions of oxidative stress and mitochondrial dysfunction. USP22 activation upon HFD feeding suggests that dysfunctional proteins may have been formed in hippocampi of HFD+sham mice, possibly as a consequence of oxidative stress, and that USP22 may help degrading these damaged proteins, a process that seems to be reduced in HFD+WATx mice. USP22's effects in the brain are however unclear, as its knockdown has shown both beneficial and detrimental outcomes in different (acute or chronic) disease models [62,63]. However, while clearly suggested by the RNAseq data, conclusions regarding USP22 inactivation need future validation by qPCR, in situ hybridization, or proteomics analyses.

An important limitation of this study is whether mouse eWAT accurately translates to abdominal visceral WAT in humans. Given that omental WAT is minimally present in mice [64], numerous similarities have been identified between mouse eWAT and human omental WAT, particularly in their roles in metabolic regulation, cholinergic regulation, and inflammation [65,66]. Both depots can become inflamed (e.g., they develop CLS formed by immune cells that surround one or more defect adipocytes) and dysfunctional (e.g., hypertrophic adipocytes that release inflammatory factors including cytokines and lipids) in the context of obesity, contributing to systemic insulin resistance and metabolic comorbidities [24,25]. However, it is important to mention that mouse eWAT and human omental WAT have different anatomical location, biological function and draining system [66]. Although both depots contribute to the development of metabolic comorbidities through systemic inflammation and circulating metabolites (e.g., release of fatty acids), omental WAT may also influence nearby metabolic organs (e.g., the liver) through additional short-distance mechanisms due to its close anatomical proximity within the abdomen [67]. Furthermore, the lipectomy performed in this study in mice (eWAT removal) should not be compared to a standard lipectomy surgery performed in humans (typically removing subcutaneous WAT and not visceral WAT). It is likely that, upon lipectomy, other compensatory mechanisms take place in humans in comparison with mice. Thus, while eWAT lipectomy in mice provides valuable proof-of-concept insights regarding the role of a WAT depot with high secretory capacity (e.g., release of cytokines, lipids) for brain health, it is difficult to translate this to the human situation, and it remains important to consider the aforementioned anatomical and functional differences when translating findings to human physiology. It is finally noteworthy to mention that eWAT, on purpose, was not entirely removed during surgery because this would cause health issues (e.g., tissue necrosis, undesired effects on testis) to the animals.

## Conclusions

In this study we demonstrated that visceral WAT is causally implicated in several aspects of obesity-related brain impairment. Without affecting body weight, food intake or metabolic risk factors, partial eWAT lipectomy alleviated HFD-induced pathological changes in cerebral vasoreactivity of the hippocampus, the main brain region involved in memory function. It also prevented the development of HFD-induced sedentary behavior, and ameliorated cortico-hippocampal functional connectivity. The beneficial effects of partial eWAT lipectomy on brain function were accompanied by a reduction in the number of very hypertrophic (abnormally enlarged) adipocytes in the residual-eWAT and changes in eWAT secretome (reduced secretion of leptin, PAI-1, MIP-1α, IL-17), suggesting that these pro-inflammatory mediators and adipokines may, at least partly, mediate the detrimental effects of obesity on cerebrovascular function. The beneficial elevations of specific plasma FFAs with anti-inflammatory and vaso-protective properties (e.g., DHA, eicosatrienoic acid) indicates that bioactive lipids may constitute important mediators between WAT and the brain. Our study demonstrates causal effects of eWAT on brain health, and therefore supports therapeutic concepts that consider direct or indirect targeting of visceral WAT dysfunction to preserve brain health and attenuate obesity-associated effects.

## Supporting information

**S1 File. Supplementary methods.**
(PDF)

**S2 File. Supplementary figures.**
(PDF)

**S3 File. Supplementary tables.**
(PDF)

## Acknowledgments

We would to thank Nicoleta Cius and Nancy Pouwels for their great help in performing the animal experiments as well as PRIME biotechnicians for taking excellent care of the mice included in this study. We also would like to acknowledge Andor Veltien for his support with the MRI, and Martien Caspers and Lars Verschuren for their help with RNAseq analyses. We also thank Aswin Menke, Nicole Worms, and Nanda Keijzer for their technical support. We finally acknowledge the technical support of the Core Facility Metabolomics and Proteomics at Helmholtz Munich for conducting the Olink experiment.

## Author contributions

**Conceptualization:** Florine Seidel, Martine C. Morrison, Ilse Arnoldussen, Maximilian Wiesmann, Robert Kleemann, Amanda J. Kiliaan.

**Formal analysis:** Florine Seidel, Martine C. Morrison, Simon Ebert, Eveline Gart, Jürgen Bernhagen, Maximilian Wiesmann, Robert Kleemann, Amanda J. Kiliaan.

**Funding acquisition:** Robert Kleemann.

**Investigation:** Florine Seidel, Vivienne Verweij, Joline Attema, Christa de Ruiter, Wim van Duyvenvoorde, Bram Geenen, Ayla Franco.

**Project administration:** Martine C. Morrison, Ilse Arnoldussen, Maximilian Wiesmann, Robert Kleemann, Amanda J. Kiliaan.

**Supervision:** Martine C. Morrison, Ilse Arnoldussen, Maximilian Wiesmann, Robert Kleemann, Amanda J. Kiliaan.

**Visualization:** Florine Seidel.

**Writing – original draft:** Florine Seidel, Martine C. Morrison, Ilse Arnoldussen, Vivienne Verweij, Simon Ebert, Joline Attema, Christa de Ruiter, Wim van Duyvenvoorde, Jessica Snabel, Bram Geenen, Ayla Franco, Eveline Gart, Jürgen Bernhagen, Maximilian Wiesmann, Robert Kleemann, Amanda J. Kiliaan.

**Writing – review & editing:** Florine Seidel, Martine C. Morrison, Ilse Arnoldussen, Vivienne Verweij, Simon Ebert, Joline Attema, Christa de Ruiter, Wim van Duyvenvoorde, Jessica Snabel, Bram Geenen, Ayla Franco, Eveline Gart, Jürgen Bernhagen, Maximilian Wiesmann, Robert Kleemann, Amanda J. Kiliaan.

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
