## [Decision Letter · Decision Letter 0]

15 Apr 2025

Dear Dr. Kiliaan,

Thank you for submitting your manuscript to PLOS ONE. After careful consideration, we feel that it has merit but does not fully meet PLOS ONE’s publication criteria as it currently stands. Therefore, we invite you to submit a revised version of the manuscript that addresses the points raised during the review process.

We look forward to receiving your revised manuscript.

Kind regards,

Peng Gao, Ph.D.

Academic Editor

PLOS ONE

Journal Requirements:

3. In the online submission form, you indicated that Datasets are available upon request and hippocampal RNAseq data will be available in the Gene Expression Omnibus (GEO) repository (https://www.ncbi.nlm.nih.gov/gds) after acceptance.

Additional Editor Comments :

The evidence supporting improved hippocampal vasoreactivity and cortico-hippocampal connectivity post-lipectomy is compelling, backed by robust MRI and behavioral data. However, the cognitive outcomes are mixed, with no clear improvement in learning tasks despite enhanced connectivity. This discrepancy suggests that other factors (e.g., aging, additional comorbidities) may modulate cognitive outcomes. The histological absence of neuroinflammation contrasts with RNAseq findings, highlighting the need for complementary techniques (e.g., multiplex cytokine assays) to resolve this inconsistency.

1. Discuss how eWAT findings might extrapolate to human visceral fat, addressing anatomical/functional differences and potential compensatory mechanisms in humans.

2. Perform additional cognitive tests (e.g., fear conditioning) to reconcile behavioral findings with imaging results. Consider longer post-surgery follow-ups to capture delayed cognitive effects.

3. Measure arteriovenous differences in adipokines/cytokines or use kinetic labeling to better quantify WAT-derived systemic contributions.

4. Reanalyze CBF data to explicitly compare pre- vs. post-surgery trends within groups, as the current interpretation of “masked aging effects” is speculative without direct statistical support.

Reviewers' comments:

Reviewer's Responses to Questions

**Comments to the Author**

1. Is the manuscript technically sound, and do the data support the conclusions?

Reviewer #1: Yes

2. Has the statistical analysis been performed appropriately and rigorously?

Reviewer #1: Yes

3. Have the authors made all data underlying the findings in their manuscript fully available?

Reviewer #1: Yes

4. Is the manuscript presented in an intelligible fashion and written in standard English?

Reviewer #1: Yes

Reviewer #1: This paper mainly studied the Partial removal of visceral epididymal white adipose tissue in obese Ldlr-/-.Leiden mice alters adipokine secretion and improves cerebrovascular health. There are some problems that need to be modified.

The main problem:

1. Please combine Figure 1 and Figure 2 in a picture and mark the words Ldlr-/- mice in Figure 1.

2. In Figure S1, only statistical diagrams are available. Please add representative histological pictures.

3. What is the control group for these values in Table 1? Please explain the result.

4. There are Figure1-9 in this article.Please modify no more than 6 Figures, for example, Figure 4 and 9 can be placed in the supplementary data.

5. In Figure 9, only sequencing results were analyzed, and no validation experiments were conducted on USP22. the conclusion "Partial eWAT lipectomy specifically inactivates USP22 in the hippocampus" lacks rigor.

**Do you want your identity to be public for this peer review?** For information about this choice, including consent withdrawal, please see our Privacy Policy

Reviewer #1: No

---

## [Author Response · Author response to Decision Letter 1]

1 Sep 2025

Dear reviewers,

We would like to thank you for the thoughtful comments and suggestions to improve our manuscript. We have provided an answer for each comment in the Word document entitled 'Response to reviewers'. The comments from the editor and reviewers are in black, followed by our answer and the corresponding changes as added in the manuscript in blue. In the revised manuscript itself, these edits appear as “tracked changes".

Yours sincerely, on behalf of all authors,

Amanda Kiliaan (corresponding author)

---

## [Decision Letter · Decision Letter 1]

9 Sep 2025

Partial removal of visceral epididymal white adipose tissue in obese Ldlr-/-.Leiden mice impacts adipokine secretion, plasma free fatty acids, and improves cerebrovascular health

PONE-D-25-13042R1

Dear Dr. Kiliaan,

We’re pleased to inform you that your manuscript has been judged scientifically suitable for publication and will be formally accepted for publication once it meets all outstanding technical requirements.

Kind regards,

Peng Gao, Ph.D.

Academic Editor

PLOS ONE

Additional Editor Comments (optional):

Reviewer #1:

Reviewers' comments:

Reviewer's Responses to Questions

**Comments to the Author**

Reviewer #1: All comments have been addressed

2. Is the manuscript technically sound, and do the data support the conclusions?

Reviewer #1: Yes

3. Has the statistical analysis been performed appropriately and rigorously?

Reviewer #1: Yes

4. Have the authors made all data underlying the findings in their manuscript fully available?

Reviewer #1: Yes

5. Is the manuscript presented in an intelligible fashion and written in standard English?

Reviewer #1: Yes

Reviewer #1: This article has undergone numerous revisions based on the specific review comments. Each review comment has been responded to rigorously, and the quality of the images is high. Therefore, I agree to accept this article.

**Do you want your identity to be public for this peer review?** For information about this choice, including consent withdrawal, please see our Privacy Policy

Reviewer #1: No

---

## [Editor Report · Acceptance letter]

PONE-D-25-13042R1

PLOS ONE

Dear Dr. Kiliaan,

I'm pleased to inform you that your manuscript has been deemed suitable for publication in PLOS ONE. Congratulations! Your manuscript is now being handed over to our production team.

Kind regards,

on behalf of

Professor Peng Gao

Academic Editor

PLOS ONE